# Interspecies interactions induce exploratory motility in *Pseudomonas aeruginosa*

Dominique H Limoli[1,2]*, Elizabeth A Warren[1], Kaitlin D Yarrington[1], Niles P Donegan[2], Ambrose L Cheung[2], George A O'Toole[2]

[1]Department of Microbiology and Immunology, Carver College of Medicine, University of Iowa, Iowa City, United States; [2]Department of Microbiology and Immunology, Geisel School of Medicine at Dartmouth, Hanover, United States

**Abstract** Microbes often live in multispecies communities where interactions among community members impact both the individual constituents and the surrounding environment. Here, we developed a system to visualize interspecies behaviors at initial encounters. By imaging two prevalent pathogens known to be coisolated from chronic illnesses, *Pseudomonas aeruginosa* and *Staphylococcus aureus*, we observed *P. aeruginosa* can modify surface motility in response to secreted factors from *S. aureus*. Upon sensing *S. aureus*, *P. aeruginosa* transitioned from collective to single-cell motility with an associated increase in speed and directedness – a behavior we refer to as 'exploratory motility'. Explorer cells moved preferentially towards *S. aureus* and invaded *S. aureus* colonies through the action of the type IV pili. These studies reveal previously undescribed motility behaviors and lend insight into how *P. aeruginosa* senses and responds to other species. Identifying strategies to harness these interactions may open avenues for new antimicrobial strategies.

*For correspondence: dominique-limoli@uiowa.edu

Competing interests: The authors declare that no competing interests exist.

## Introduction

While it is clear that many microbial infections do not occur with a single species, we have only recently begun to understand the profound impacts microbial species have on each other and patients (*Nguyen and Oglesby-Sherrouse, 2016*). Studies of dental biofilms, intestinal communities, chronic wounds, and respiratory infections in patients with cystic fibrosis (CF) demonstrate that community interactions influence microbial survival and disease progression (*Limoli and Hoffman, 2019*; *Gabrilska and Rumbaugh, 2015*). For example, we and others find an association between coisolation of *Pseudomonas aeruginosa* and *Staphylococcus aureus* from the CF airway or chronic wounds and poor patient outcomes, including decreased lung function and shortened life-spans (*Limoli et al., 2016*; *Maliniak et al., 2016*; *Hubert et al., 2013*). Laboratory studies also reveal interactions between these two pathogens can alter virulence factor production by one or both species, potentially influencing pathogenesis, persistence, and/or antibiotic susceptibility (*Hotterbeekx et al., 2017*). For example, in a model of coinfection on CF-derived bronchial epithelial cells, we observed *P. aeruginosa* and *S. aureus* form mixed microcolonies, which promotes the survival of *S. aureus* in the presence of vancomycin (*Orazi and O'Toole, 2017*). One strategy to improve outcomes for these coinfected patients may be to block harmful interspecies interactions before they begin.

Here, we designed a system to visualize early interactions between *P. aeruginosa* and *S. aureus* and follow single-cell behaviors over time with live-imaging. We show that *P. aeruginosa* can sense *S. aureus* secreted products from a distance and in turn, dramatically alter the motility behaviors of this Gram-negative bacterium. In response to *S. aureus*, individual *P. aeruginosa* cells transition from

collective to single-cell movement, allowing exploration of the surrounding environment and directional movement towards *S. aureus.* We find that such 'exploratory motility' is driven primarily by the *P. aeruginosa* type IV pili (TFP) and modulated, in part, by the CheY-like response regulator, PilG. Importantly, we find *P. aeruginosa* is capable of responding to an array of bacterial species and strains recovered from a variety of sources, including CF patients, demonstrating a broad capacity for *P. aeruginosa* to sense other microbial species and modulate motility in response. Thus, we provide a new means to study polymicrobial interactions at the single-cell level and reveal that *P. aeruginosa* can sense the presence of other microbial species and dramatically, yet specifically, modify its behavior in response to such interspecies signals.

## Results

### *S. aureus* promotes exploratory motility in *P. aeruginosa*

To understand early microbial interactions, we established an in vitro coculture system to monitor *P. aeruginosa* and *S. aureus* at first encounters. Bacteria were inoculated at low cell densities between a coverslip and an agarose pad in minimal medium, supplemented with glucose and tryptone, and imaged with phase contrast time-lapse microscopy every 15 min for 8 hr. Alone, *P. aeruginosa* cells replicate and expand outward as raft-like groups, as previously described for *P. aeruginosa* surface-based motility (*Anyan et al., 2014*; *Burrows, 2012*) (*Video 1*; *Figure 1*, still montage, top row). In comparison, coincubation with *S. aureus* resulted in dramatically altered behavior (*Video 2*, *Figure 1*, still montage, bottom row). After two to three rounds of cell division, instead of remaining as a group, individual *P. aeruginosa* cells began to increase motility and moved as single-cells, suggesting that *P. aeruginosa* responds to the presence of *S. aureus* by altering motility behaviors. *P. aeruginosa* significantly inhibited *S. aureus* growth, as previously reported (see *Figure 1—video 1* for representative time-lapse video of *S. aureus* alone).

To visualize *P. aeruginosa* motility in the presence of *S. aureus* in more detail, the inoculating cell density was reduced (2–3 cells of each species per field of view), and images were taken at 5 s intervals for 8 hr. *Video 3* shows images taken during hours 4–6 of coculture, when *P. aeruginosa* initiates single-cell movement under these conditions (*Figure 2A*, still montage). We observed a number of surprising behaviors by *P. aeruginosa* in the presence of *S. aureus. P. aeruginosa* cells initially replicated and remained in a raft (t = 4 h:28 m), as we and others have observed for *P. aeruginosa* in monoculture, but as the community approached *S. aureus,* individual cells: (1) exited the raft (t = 4 h:34 m), (2) moved with increased speed, and (3) moved towards and surrounded *S. aureus,* 'explored' the surface of the colony until (*Figure 2A–B*), (4) *P. aeruginosa* cells entered the *S. aureus* colony (*Figure 2C*, t = 6 hr), and finally, (5) by 8 hr, *P. aeruginosa* completely dismantled the *S. aureus* community (*Figure 2D*, t = 8 hr). *P. aeruginosa* was also observed to adopt a swift motion, beginning between 4.5–6 hr, moving in and out of the plane of focus during imaging (*Video 4*, yellow circle and *Figure 2D*, red arrows).

To quantitate the movement of *P. aeruginosa* single-cell motility in comparison to collective motility in rafts, the movement of individual cells and the leading edge of the rafts was tracked over time. In comparison to cells moving in rafts, individual cells moved with increased speed (μm/s), acceleration (μm/s$^2$), and mean squared displacement (MSD, μm$^2$) (*Figure 2E, F and G*, respectively). MSD represents a combined measure of both the speed and directional persistence of the cell, thus an increased MSD in single-cells over a change in time suggests single-cells exhibit directed motion, followed by a decreased MSD when *P. aeruginosa* cells reach the *S. aureus* colony.

### *P. aeruginosa* type IV pili drive exploratory motility

How is *P. aeruginosa* motility generated in response to *S. aureus*? When grown in monoculture, *P. aeruginosa* performs cellular movement through the action of a single polar flagellum and the type IV pili (TFP) (*Conrad et al., 2011*; *Gibiansky et al., 2010*; *Merritt et al., 2010*). To determine how the observed *P. aeruginosa* 'exploratory motility' is generated, *P. aeruginosa* strains deficient in the production of either TFP (ΔpilA; encoding the pilin monomers), flagella (ΔflgK; encoding the flagella hook protein), or both (ΔpilA ΔflgK) were examined. Time-lapse images were taken as described for *Figure 2*, except they were acquired at 50 ms intervals for visualization of specific motility patterns. Representative videos and snap-shots (*Figure 3A*) were chosen at the time-point where *P.*

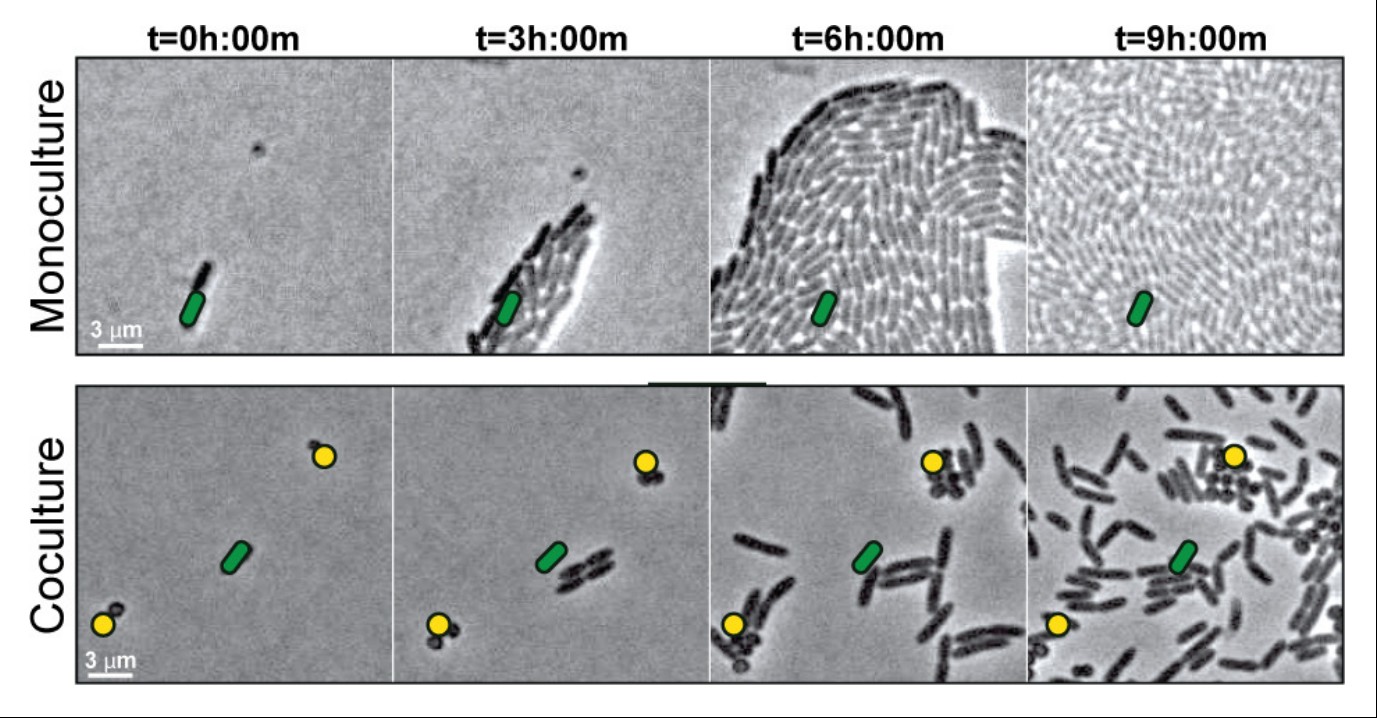

**Figure 1.** *S. aureus* increases *P. aeruginosa* motility. Live-imaging of polymicrobial interactions. *P. aeruginosa* (rod-shaped) was inoculated between a coverslip and an agarose pad, either in monoculture (top) or in coculture with equal numbers of *S. aureus* (cocci-shaped, bottom). Images were acquired every 15 m for 9 hr. Representative snap-shots of *Video 1* (top) and *Video 2* (bottom) are shown. Founding cells identified in the first frame are indicated with green rods (*P. aeruginosa*) or yellow circles (*S. aureus*). The location of the founding cell is indicated in each subsequent frame for positional reference. At t = 9 h:00 m, the founding *P. aeruginosa* cell has moved outside the field of view.

The online version of this article includes the following video for figure 1:

**Figure 1—video 1.** WT *S. aureus* in monoculture.

https://elifesciences.org/articles/47365#fig1video1

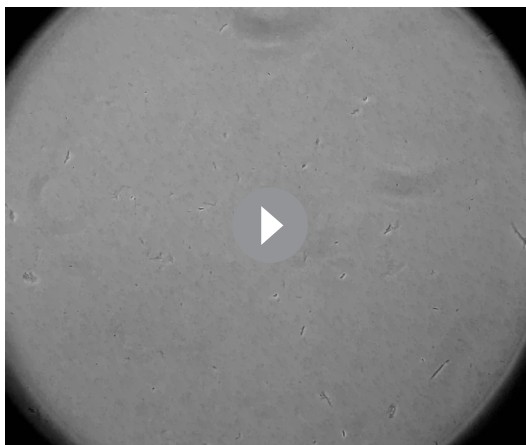

**Video 1.** WT *P. aeruginosa* in monoculture. Duration 8 hr. Acquisition interval 15 m. Playback speed 3054x.

https://elifesciences.org/articles/47365#video1

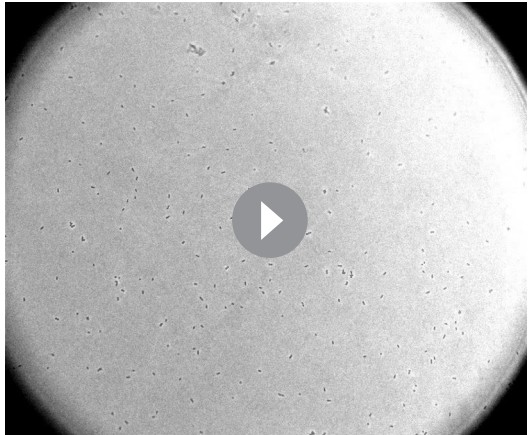

**Video 2.** WT *P. aeruginosa* in coculture with WT *S. aureus*. Duration 8 hr. Acquisition interval 15 m. Playback speed 3054x.

https://elifesciences.org/articles/47365#video2

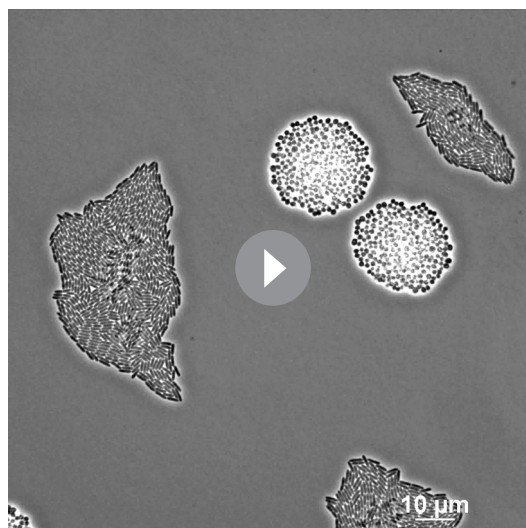

**Video 3.** WT *P. aeruginosa* in coculture with WT *S. aureus*. Duration 10 m. 4 hr post inoculation. Acquisition interval 5 s. Playback speed 50x.
https://elifesciences.org/articles/47365#video3

*aeruginosa* was found to exhibit both slow and swift single-cell movements. The Δ*pilA* mutant was unable to move away from the group as single-cells (*Video 5*), as seen for the parental *P. aeruginosa* (*Video 4*), suggesting the TFP are required for *P. aeruginosa* exploratory motility. However, the swift movement observed in the WT (*Figure 3A*, see boxed inset with red arrows), was maintained in Δ*pilA*, demonstrating that TFP are not required for this behavior.

We next examined the Δ*flgK* mutant (*Figure 3—video 1*), which was seen to adopt the single-cell behaviors of the WT, except the swift movements were not observed, supporting the hypothesis that this movement is generated by the flagellum. These data suggest that while *S. aureus* modulates both TFP and flagella-mediated motility, the early events necessary for exploration (initiation of single-cell movement and directional movement towards *S. aureus*) require the TFP.

We also examined the response of a double Δ*pilA* Δ*flgK* mutant in the presence of *S. aureus* (*Figure 3—video 2*). Surprisingly, this mutant exhibited a phenotype distinct from either the WT, or the individual Δ*pilA* and Δ*flgK* mutants. *P. aeruginosa* cells deficient in both TFP and flagella were not only unable to produce the respective movements characteristic of these motility motors, but also were unable to remain within a raft-like group (*Figure 3A*). This behavior was not dependent upon the presence of *S. aureus*, as a similar pattern was observed for Δ*pilA* Δ*flgK* when visualized in the absence of *S. aureus* (*Figure 3—figure supplement 1*).

The influence of *P. aeruginosa* exploratory motility on *S. aureus* growth was also examined by measuring the area of the *S. aureus* colonies at 4.5 hr post-inoculation. *S. aureus* colonies were significantly smaller in the presence of WT *P. aeruginosa* or the Δ*flgK* mutant in comparison to *S. aureus* grown in monoculture (*Figure 3B*). However, in the presence the Δ*pilA* mutant, which was deficient in exploratory motility, colonies were significantly larger in comparison to WT and the Δ*flgK* mutant. Moreover, when grown in the presence of the double mutant, the area of the *S. aureus* colonies was not significantly different from *S. aureus* monoculture. These data suggest that ability of *P. aeruginosa* to perform exploratory motility influences *P. aeruginosa* inhibition of *S. aureus* growth.

While visualizing *P. aeruginosa-S. aureus* interactions at 50 ms intervals, we observed an additional *P. aeruginosa* behavior. When *P. aeruginosa* first encounters the *S. aureus* colony, the cell body orients perpendicular to the surface of the colony and moves back-and-forth (*Video 4*, blue circle and *Figure 3C*) and appears to move *P. aeruginosa* cells into the *S. aureus* colony. This behavior was only observed in the WT and Δ*flgK* mutant, suggesting that it is driven by the TFP.

## Agr-regulated secreted *S. aureus* factors promote *P. aeruginosa* motility

Live-imaging suggests *P. aeruginosa* is capable of sensing *S. aureus* and initiating exploratory motility from a distance. Thus, we hypothesized that *P. aeruginosa* responds to *S. aureus* secreted factors. Since we observed that TFP were required for exploratory motility, we sought to develop a macroscopic assay where TFP motility could be monitored in the presence of *S. aureus* secreted factors only. Macroscopically, population-scale TFP-mediated twitching motility can be visualized by inoculating *P. aeruginosa* cells onto the bottom of a plate through 1.5% agar (sub-surface); cells move on the surface of the plate under the agar and can be stained with crystal violet for visualization (*Turnbull and Whitchurch, 2014*). To test the hypothesis that *P. aeruginosa* responds to *S. aureus* secreted products, cell-free supernatant derived from overnight cultures of WT *S. aureus*, (normalized to $OD_{600}$ = 5.0) was spread on the plate, prior to pouring molten agar. *S. aureus* supernatant

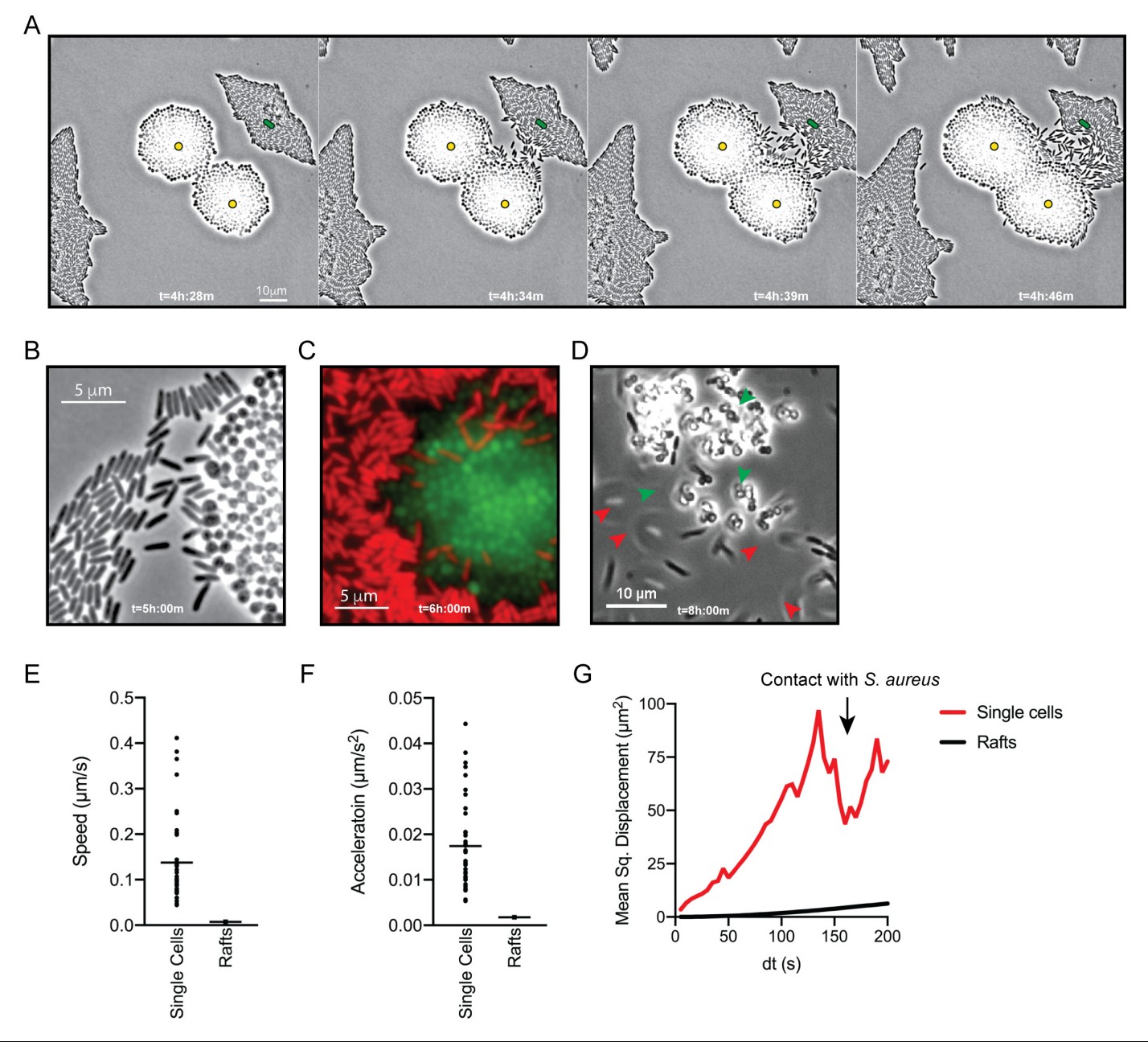

**Figure 2.** *P. aeruginosa* adopts an exploratory mode of motility in the presence of *S. aureus*. Live-imaging of *P. aeruginosa* with WT *S. aureus* (*Video 3*). (**A**) Montage of representative snap-shots are shown beginning at 4 hr:28 m. Founding cells identified in the first frame are indicated with green rods (*P. aeruginosa*) or yellow circles (*S. aureus*). The location of the founding cell is indicated in each subsequent frame for positional reference. (**B**) Snap-shot at 5 hr, zoomed in to visualize single-cells. (**C**) Snap-shot at 6 hr of *P. aeruginosa* (mKO, red) and *S. aureus* (GFP, green) illustrating *P. aeruginosa* invasion into *S. aureus* colonies. (**D**) Snap-shot of coculture at 8 hr, showing disruption of *S. aureus* colonies (green arrows) and swift-moving *P. aeruginosa* cells out of the plane of focus (red arrows). Single *P. aeruginosa* cells and the leading edge of rafts were tracked in the presence of *S. aureus* and the speed (μm/s), acceleration (μm/s²), and mean squared displacement (μm²) for four independent videos are indicated, respectively, in (**E** – **G**).

significantly increased the motility diameter of *P. aeruginosa* in a dose-dependent manner (*Figure 4A*), supporting the hypothesis that *S. aureus* secreted factors increase *P. aeruginosa* twitching motility.

Two primary regulators of secreted factors in *S. aureus* are the alternative stress sigma factor, sigma B (SigB) (*Bæk et al., 2013*) and the accessory gene regulator (Agr) quorum sensing system

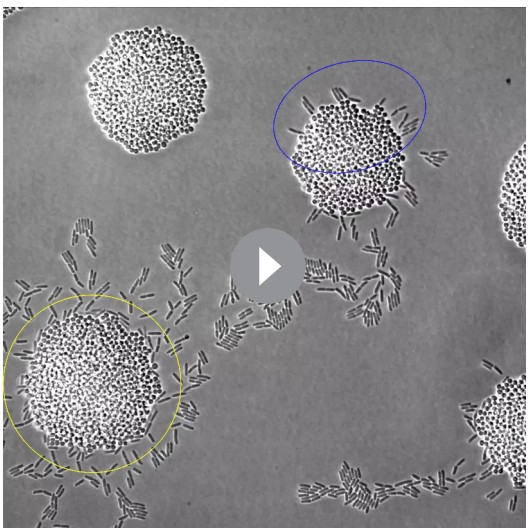

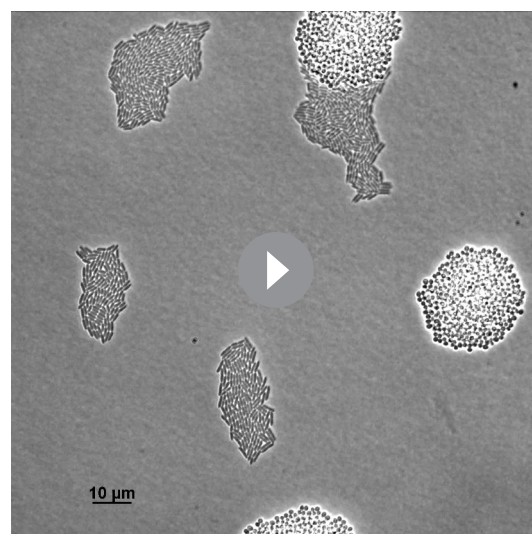

**Video 4.** WT *P. aeruginosa* in coculture with WT *S. aureus*. Duration 30 s. 4.5 hr post inoculation. Acquisition interval 50 ms. Playback speed 3x. https://elifesciences.org/articles/47365#video4

**Video 5.** *P. aeruginosa* Δ*pilA* in coculture with WT *S. aureus*. Duration 30 s. 4.5 hr post inoculation. Acquisition interval 50 ms. Playback speed 3x. https://elifesciences.org/articles/47365#video5

(*Nair et al., 2011*). *S. aureus* strains with transposon insertions in either *sigB* or *agrB* (*Fey et al., 2013*) were examined for their ability to induce *P. aeruginosa* twitching motility (*Figure 4B*). While the *sigB::*Tn strain phenocopied the WT, the *agrB::*Tn mutant lost all ability to induce *P. aeruginosa* twitching motility, suggesting that Agr regulates the production of the factors promoting motility in *P. aeruginosa*. The *agr* operon is organized around two divergent promoters, P2 and P3, and generates two primary transcripts, RNAII and RNAIII, respectively. RNAII encodes AgrB, AgrD, AgrC, and AgrA (*Le and Otto, 2015*). To confirm a role for the Agr quorum sensing system, an unmarked deletion of *agrBDCA* was generated and complemented with WT *agrBDCA. P. aeruginosa* motility was examined in the presence of supernatant derived from these strains. As predicted, the Δ*agrBDCA* mutant was unable to enhance *P. aeruginosa* motility, and complementation restored activity to WT levels (*Figure 4C*).

To confirm that the TFP are necessary for increased *P. aeruginosa* motility in this assay, we examined the response of *pilA*-deficient *P. aeruginosa* to *S. aureus* supernatant. In the absence of *S. aureus* supernatant, the Δ*pilA* mutant was unable to perform twitching motility, as previously reported (*Darzins, 1994*). However, while motility was reduced in the presence of *S. aureus* supernatant, surprisingly, the Δ*pilA* mutant retained some ability to respond to *S. aureus* secreted factors (*Figure 4D*). Since we observed during live-imaging that *S. aureus* can increase *P. aeruginosa* flagella-mediated motility, in addition to TFP-mediated motility, we hypothesized that the response retained in the Δ*pilA* mutant was due to increased flagella-mediated motility. To test this hypothesis, we examined the response of Δ*flgK* and a double Δ*pilA* Δ*flgK* mutant to *S. aureus* supernatant. The single Δ*flgK* mutant phenotype trended lower, but was not significantly different from WT, while the motility of the Δ*pilA* Δ*flgK* mutant was reduced to levels not significantly different from Δ*pilA* or Δ*pilA* Δ*flgK* in the absence of supernatant (*Figure 4D*). These data support our observation from live-imaging that, while TFP were the primary contributors to increased motility, *S. aureus* secreted factors could increase both TFP- and flagella-mediated motility.

To formally examine the response of *P. aeruginosa* flagella-mediated motility to *S. aureus* supernatant, traditional macroscopic low-percentage agar assays that measure the contribution of flagella motility and chemotaxis were performed. Cell-free *S. aureus* supernatant was mixed into 0.3% agar prior to inoculating *P. aeruginosa* cells into the agar. The diameter of the *P. aeruginosa* motility zone showed a modest, but not significant increase in the presence of *S. aureus* supernatant, compared to medium alone (*Figure 4E*). To examine the role for flagella in response to *S. aureus* under swim assay conditions, the Δ*flgK* mutant was tested. As expected, flagella-mediated motility, both with

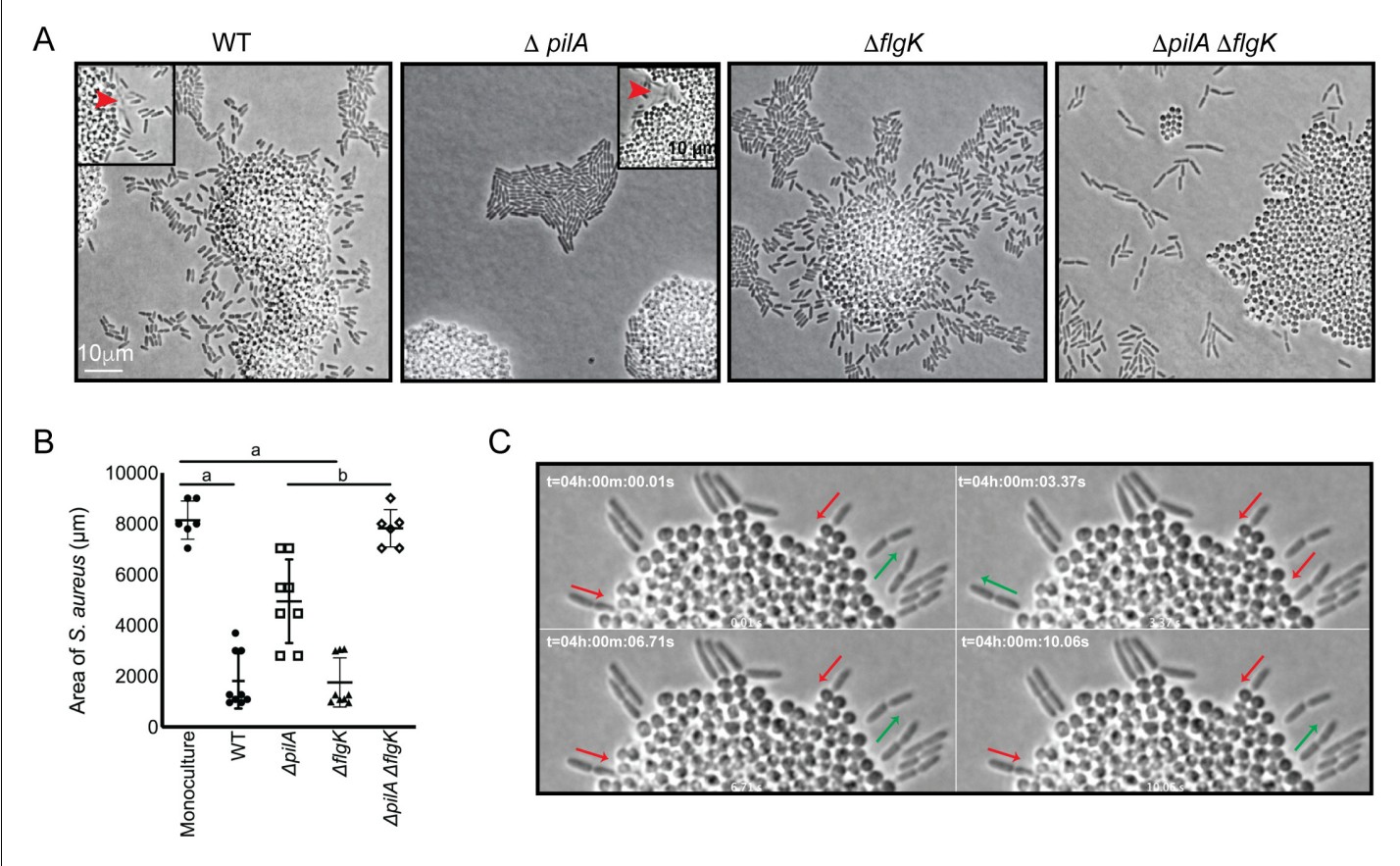

**Figure 3.** *P. aeruginosa* exploratory motility is driven by type IV pili. Live-imaging of *P. aeruginosa* with WT *S. aureus*. (**A**) Representative snap-shots of coculture with WT *S. aureus* and *P. aeruginosa* (WT, Δ*pilA*, Δ*flgK,* and Δ*pilA* Δ*flgK*, left to right, *Videos 4* and *5* and *Figure 3—videos 1* and *2* respectively) are shown at t = 4.5 hr. Boxed insets show swift-moving *P. aeruginosa* cells out of the plane of focus (red arrows). (**B**) The area of *S. aureus* per frame in monoculture or in the presence of the indicated *P. aeruginosa* strain was calculated at t = 5 hr by dividing the total area occupied by *S. aureus* in a single frame by the number of *S. aureus* colonies. A minimum of four videos were analyzed per condition. The mean and standard deviation are indicated. Statistical significance was determined by one-way ANOVA followed by Tukey's Multiple Comparisons Test - *a* indicates a statistically significant difference (p≤0.05) between *S. aureus* in monoculture and in the presence of either WT *P. aeruginosa* or Δ*flgK*; *b* indicates a statistically significant difference between Δ*pilA* and Δ*pilA* Δ*flgK*. (**C**) Representative snap-shots of *Video 4*, WT *P. aeruginosa* and *S. aureus* beginning at 4 hr with 50 ms intervals, showing back-and-forth motion. Red arrows indicate when a *P. aeruginosa* cell is moving in towards the *S. aureus* colony and green arrows indicate when a cell is moving away.

The online version of this article includes the following video and figure supplement(s) for figure 3:

**Figure supplement 1.** Live-imaging of *P. aeruginosa* Δ*pilA* Δ*flgK* mutant in monoculture.

**Figure 3—video 1.** *P. aeruginosa* Δ*flgK* in coculture with WT *S. aureus*.
https://elifesciences.org/articles/47365#fig3video1

**Figure 3—video 2.** *P. aeruginosa* Δ*pilA* Δ*flgK* in coculture with WT *S. aureus*.
https://elifesciences.org/articles/47365#fig3video2

and without *S. aureus* supernatant, was significantly reduced. To determine if TFP contribute under these assay conditions, the Δ*pilA* and Δ*pilA* Δ*flgK* mutants were also examined. The motility diameter of Δ*pilA* was not significantly different from WT, and the double mutant phenocopied the single Δ*flgK* mutant, suggesting that flagella are primarily responsible for the motility observed here.

## *P. aeruginosa* biases the directionality of movement up a concentration gradient of *S. aureus*-secreted factors

Plate-based macroscopic motility assays measure both an absolute increase in motility and directional movement up a self-generated gradient (chemotaxis) as bacterial populations metabolize the available substrates and expand outward radially (*Shapiro, 1984*). While flagella-based movement

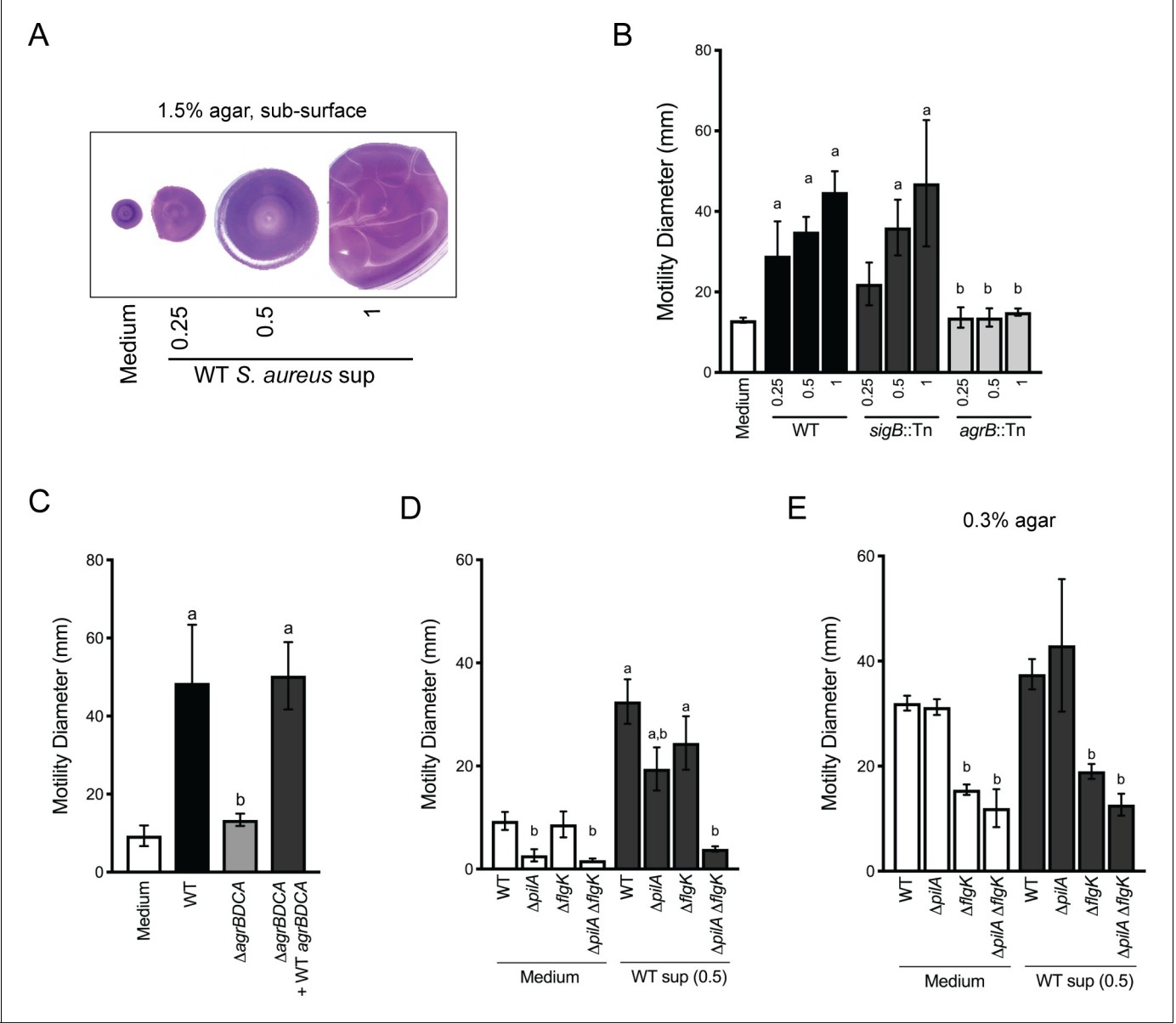

**Figure 4.** Agr-regulated secreted *S. aureus* factors increase *P. aeruginosa* motility. (A – D) Motility of WT *P. aeruginosa* was monitored by macroscopic sub-surface inoculation assays in the presence of medium alone or cell-free supernatant from the indicated *S. aureus* strains. (A) illustrates representative motility zones stained with crystal violet for visualization. The dilution factor of the supernatant is indicated in **A** and **B**. Undiluted supernatant was used in **C**. In **D** and **E**, the motility of the indicated *P. aeruginosa* mutants was analyzed in the presence of medium alone or supernatant derived from WT *S. aureus* (0.5 dilution) under 1.5% agar (**D**) or within 0.3% agar (**E**). The mean and standard deviation are indicated for at least three biological replicates. Statistical significance was determined by one-way ANOVA followed by Tukey's Multiple Comparisons Test - *a* indicates a statistically significant difference (p≤0.05) between the motility observed in the presence of *S. aureus* supernatant compared to medium alone, and *b* indicates a statistically significant difference (p≤0.05) between the motility observed in the mutant strain (*S. aureus* mutants in **B** and **C**, *P. aeruginosa* mutants in **D** and **E**) compared to the parental.

through liquid has been extensively described in *P. aeruginosa* for a variety of chemoattractants, directional movement on a surface is poorly understood. Kearns et al. previously reported *P. aeruginosa* pili-mediated biased movement up a gradient of phosphatidylethanolamine (PE) on the surface of an agar plate (*Kearns et al., 2001*). To determine if *P. aeruginosa* is capable of moving up a previously established concentration gradient of *S. aureus* supernatant, supernatant derived from either WT or the Δ*agrBDCA* mutant of *S. aureus* was spotted onto the surface of 1.5% agar (containing buffered medium only, no carbon source). The secreted factors were allowed to diffuse and establish

a gradient for approximately 24 hr, prior to inoculating *P. aeruginosa* onto the surface, as previously described (*Kearns et al., 2001*). Preferential movement of *P. aeruginosa* towards supernatant derived from WT *S. aureus* was observed, but not for medium alone or supernatant derived from the Δ*agrBDCA* mutant (*Figure 5A*). The response for the Δ*pilA* mutant was also examined and all motility was abrogated in this mutant, demonstrating motility observed in this assay is entirely pili-mediated. These data support the hypothesis that *P. aeruginosa* biases the directionality of TFP-mediated motility up a concentration gradient of *S. aureus* secreted factors.

We next asked if *P. aeruginosa* would also fail to migrate towards *S. aureus* deficient in Agr activity at the single-cell level. WT *P. aeruginosa* was visualized in coculture with Δ*agrBDCA*, as previously performed in the presence of WT *S. aureus* – with images acquired every 5 s for 8 hr (*Video 3* for WT: *Figure 2A* and *Video 6* for the Δ*agrBDCA* mutant: *Figure 5B*). In comparison to WT *S. aureus*, *P. aeruginosa* behavior was significantly altered in the presence of Δ*agrBDCA*. *P. aeruginosa* remained capable of initiating single-cell movement; however, once cells initiated single-cell movement, the path of their movement did not appear as directed towards the *S. aureus* colonies. In fact, some cells seemed to actively avoid the *S. aureus* colony all together.

To quantify the directedness of *P. aeruginosa* movement in the presence of WT and the Δ*agrBDCA* mutant, single *P. aeruginosa* cells were tracked from the first frame a cell exited the raft to the frame where it first encounters *S. aureus* (*Figure 5B*). The accumulated track distance, $D_{(A)}$, was measured and compared to the Euclidean distance, $D_{(E)}$, between the position of the cell in the first and last frame tracked. The directness of *P. aeruginosa* towards *S. aureus* was calculated as a ratio of $D_{(E)}/D_{(A)}$. *P. aeruginosa* exhibited a significantly higher directedness ratio towards WT *S. aureus* compared to the Δ*agrBDCA* mutant (*Figure 5C*). These data support the hypothesis that Agr regulates the production of *S. aureus* secreted factors driving the directionality of *P. aeruginosa* motility.

## The CheY-like response regulator, PilG, modulates *P. aeruginosa* response to *S. aureus*

How does *P. aeruginosa* sense the presence of *S. aureus* secreted products and initiate TFP-driven exploratory motility? *P. aeruginosa* encodes three known chemotaxis pathways: two flagella-mediated (*che* and *che2*) and a putative TFP-mediated system (*pil-chp*) (*Darzins, 1994*; *Whitchurch et al., 2004*). While several proteins encoded in the Pil-Chp pathway comprise a signal transduction pathway very similar (by gene homology) to the flagella-mediated chemotaxis pathway, their role in directional twitching motility remains unclear. A significant challenge lies in that many of these proteins are required for pilus assembly, thus teasing apart their requirement for motility per se versus regulation of a chemotactic response is challenging. Nonetheless, *pilJ* is predicted to encode the TFP methyl-accepting chemoreceptor protein (MCP) and by homology to the flagella MCPs, PilJ is expected to detect changes in the concentration of an attractant or repellent and initiate a signaling cascade to control twitching motility. To determine if PilJ is necessary for *P. aeruginosa* to sense *S. aureus*, we examined the response of a Δ*pilJ* mutant to *S. aureus* (WT) in the macroscopic sub-surface inoculation assay and by microscopic live-imaging. In the macroscopic assay, the Δ*pilJ* mutant exhibited low twitching motility, as previously described (*Darzins, 1994*) and responded to *S. aureus* supernatant to a similar degree as the Δ*pilA* mutant (*Figure 6—figure supplement 1*). However, examination by live-imaging revealed that Δ*pilJ* remained capable of performing surface motility and phenocopied the behavior of WT *P. aeruginosa* in the presence of *S. aureus*; moving as single-cells and with preferred directionality towards *S. aureus* (*Figure 6A*), suggesting that PilJ is not necessary for exploratory motility.

A second protein encoded within the Pil-Chp chemosensory system that has been implicated in TFP-mediated chemotaxis is the CheY-like response regulator PilG, which regulates the ATPase necessary for TFP extension (PilB). Prior work demonstrates that while a Δ*pilG* mutant is non-motile in a macroscopic sub-surface inoculation assay (as previously described, *Bertrand et al., 2010*), in a microfluidic assay, individual surface attached *P. aeruginosa* cells were capable of performing twitching motility on a single-cell level; yet, they were unable to respond to a chemotactic gradient (*Oliveira et al., 2016*). We therefore examined a role for PilG in *P. aeruginosa* response to *S. aureus*. Similar to Δ*pilJ* we found that Δ*pilG* phenocopied the Δ*pilA* mutant in the macroscopic assay and was unable to generate twitching motility (*Figure 6—figure supplement 1*). However, during live-imaging, in comparison to Δ*pilJ*, Δ*pilG* was diminished in its capacity to increase motility in response

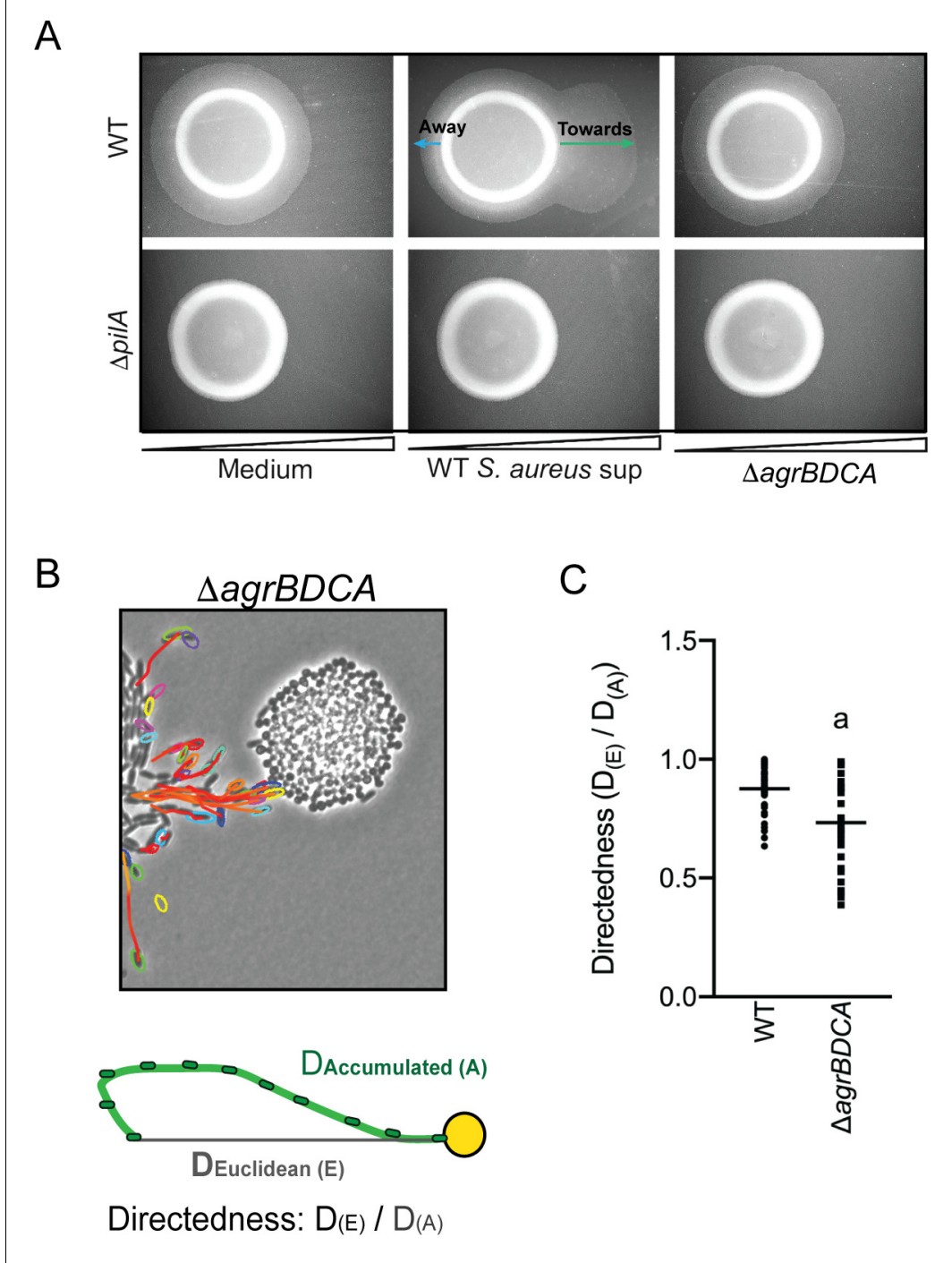

**Figure 5.** *P. aeruginosa* biases the directionality of movement up a concentration gradient of Agr-regulated secreted factors. (**A**) A concentration gradient of either *S. aureus* growth medium (TSB), *S. aureus* supernatant derived from WT or Δ*agrBDCA* was established by spotting onto the surface of the agar and allowing a concentration gradient to establish by diffusion for approximately 24 hr, prior to spotting *P. aeruginosa* onto the agar (6 mm to the left) and surface-based motility imaged after 24 hr. Representative images of at least three independent experiments are shown. (**B**) Example of live-imaging of WT *P. aeruginosa* with *S. aureus* Δ*agrBDCA* with tracks of single-cells shown and schematic illustrating the methods for calculating the directedness. Single *P. aeruginosa* cells were tracked from first frame a cell exited the raft to the frame where it first encounters *S. aureus*. The accumulated track distance, $D_{(A)}$, was measured for at least 30 cells in four independent videos and compared to the Euclidean distance, $D_{(E)}$, between the position of the cell in the first and last frame tracked. The

*Figure 5 continued on next page*

*Figure 5 continued*
ratio of D$_{(E)}$/D$_{(A)}$ (Directedness) is shown in (C) for *P. aeruginosa* moving towards WT *S. aureus* compared to Δ*agrBDCA* with the mean indicated. Statistical significance was determined by an unpaired Student's *t*-test (a = $P \leq 0.05$).

to *S. aureus* (*Figure 6B* for PA14 and *Figure 6—figure supplement 2* for PAO1, see discussion). Interestingly, we did note that in a few circumstances (3 of 9 videos), when a sufficient number of *S. aureus* cells were present (five or more founding cells per field of view) Δ*pilG* would eventually respond to the presence of *S. aureus* (*Figure 6—figure supplement 3*).

To confirm that Δ*pilG* exhibits a diminished response to *S. aureus* in comparison to WT, we imaged both WT *P. aeruginosa* and the isogeneic Δ*pilG* mutant in coculture with *S. aureus* simultaneously to account for variability in the number of *S. aureus* cells in the initial inoculum. *P. aeruginosa* strains were engineered to differentially express GFP (WT) or mKate (red, Δ*pilG*). Here the Δ*pilG* mutant displayed a diminished response to *S. aureus* in comparison to WT *P. aeruginosa* (*Figure 6C*). These data support the previous observations suggesting that a Δ*pilG* mutant is capable of performing TFP-mediated motility, but has a diminished response to chemotactic signals (*Oliveira et al., 2016*).

## *P. aeruginosa* responds to a broad range of model organisms and CF pathogens

Our data thus far demonstrate that *P. aeruginosa* can sense *S. aureus* from a distance and modulate both flagella and TFP-mediated motility. To begin to examine if these behaviors might be relevant during airway infection in CF patients, we examined the capacity of a panel of *P. aeruginosa* isolates from CF patients to respond to *S. aureus*. Phenotypic loss of twitching motility in monoculture is often observed in chronic *P. aeruginosa* isolates from CF patients (*Mayer-Hamblett et al., 2014*), thus it was not surprising that two of the four isolates examined exhibited no detectable twitching motility in the absence of *S. aureus*. Nonetheless, each *P. aeruginosa* isolate was able to respond to *S. aureus* supernatant, although to varying degrees (*Figure 7A*). Importantly, even mucoid *P. aeruginosa* isolates (CFPBPA38, 43, 37), exhibited at least a two-fold increase in motility in the presence of *S. aureus* supernatant, above medium alone.

Next, we asked if *S. aureus* isolates from CF patients are capable of inducing *P. aeruginosa* motility in the macroscopic twitching assay. Supernatant derived from two out of three clinical CF *S. aureus* isolates (each from different patients) was capable of inducing motility of the WT *P. aeruginosa* laboratory strain, to an extent not significantly different from that previously observed with WT *S. aureus* (*Figure 7B*). Since we previously observed Agr was necessary for *S. aureus* to promote motility, we hypothesized that CFBRSA48 was unable to induce *P. aeruginosa* due to reduced activity of the Agr quorum sensing system – an adaptation previously reported for *S. aureus* CF isolates (*Nair et al., 2011*; *Goerke and Wolz, 2010*). Since Agr also positively regulates the production of *S. aureus* β-hemolysin, hemolysis on sheep blood agar plates is often used as an indicator of Agr activity (*Peng et al., 1988*). However, all three clinical isolates maintain WT levels of hemolysis, suggesting Agr is active in these strains and an unknown mechanism accounts for reduced activity towards *S. aureus* in CFBRSA isolate 48 (*Figure 7—figure supplement 1*).

Can *P. aeruginosa* also respond to other CF pathogens, or is this behavior specific to *S. aureus*? To investigate this question, we examined clinical isolates from CF patients of three additional CF pathogens: *Haemophilus influenzae*, *Burkholderia cepacia*, and *Achromobacter xylosoxidans*. When possible, we examined multiple isolates for the ability of cell-free supernatant to enhance twitching motility in *P. aeruginosa*. We observed that one of two *B. cepacia* isolates tested enhanced *P. aeruginosa* motility to a similar degree as *S. aureus*, while none of those tested from *H. influenzae* or *A. xylosoxidans* significantly increased motility under these conditions (*Figure 7C*).

We then asked if *P. aeruginosa* is able to sense and respond more broadly to the presence of other bacterial species. Here we tested three additional non-CF organisms: *Bacillus subtilis*, *Escherichia coli*, and *Salmonella* Typhimurium. While the amount of *P. aeruginosa* motility observed varied between species and laboratory strains utilized, we observed a significant increase in *P. aeruginosa* motility in response to each organism tested, demonstrating that *P. aeruginosa* responds to a variety of bacterial species (*Figure 7D*).

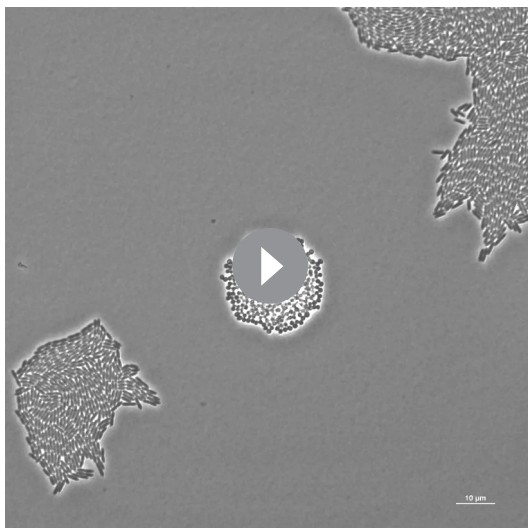

**Video 6.** WT *P. aeruginosa* in coculture with *S. aureus* Δ*agrBDCA*. Duration 10 m. 4 hr post inoculation. Acquisition interval 5 s. Playback speed 50x.
https://elifesciences.org/articles/47365#video6

To gain insight into the secreted microbial factors that promote *P. aeruginosa* motility, we performed preliminary biochemical analysis on *S. aureus* supernatant. First, cell-free supernatant was subjected to heat treatment at 95°C for 30 min. Heat treatment did not influence supernatant motility-inducing activity. However, protease treatment of the supernatant, with either proteinase k or trypsin, almost entirely eliminated activity (*Figure 8A*). These data suggest that the active factors are heat-stable proteins. Supporting this conclusion, we also found that the active factors were soluble in the aqueous fraction following hydrophobic extraction (*Figure 8B*). These data suggest the *S. aureus* factors responsible for promoting *P. aeruginosa* surface motility are proteinaceous in nature, but the activity is insensitive to denaturation by heat.

Together our biochemical and genetic evidence predict that heat-stable, Agr-regulated proteins and/or peptides promote *P. aeruginosa* motility. *S. aureus* produces such peptides referred to as phenol-soluble modulins (PSM). PSMs are amphipathic, multifunctional peptides that lyse many human cell types, are pro-inflammatory and chemoattractant to neutrophils, and influence *S. aureus* biofilm morphology and dispersal (*Cheung et al., 2014*). *S. aureus* produces five α-peptides (PSMα1–4 and δ-toxin) and two β-peptides (PSMβ1 and 2). PSMα are encoded by the *psmα* operon, PSMβ by the *psmβ* operon, and the δ-toxin (also called δ-hemolysin, Hld) is encoded within the RNAIII, regulatory RNA encoded from the *agr* operon. To determine if *S. aureus* PSMs might influence *P. aeruginosa* twitching motility, we collected supernatant from *S. aureus* deficient in the production of all seven PSMs, Δ*psmα*1–4 Δ*psm*β1–2 and δ-toxin start (ATG) to stop (ATT) mutation, to preserve expression of RNAIII (referred to as Δ*psmαβδ*, from this point forward) (*Syed et al., 2015*). Supernatant derived from the Δ*psmαβδ* mutant showed a significantly reduced ability to promote *P. aeruginosa* motility in comparison to WT; however, the activity remained higher than Δ*agrBDCA*, suggesting one or more additional unidentified *S. aureus* factors also promote *P. aeruginosa* motility (*Figure 8C*).

## Discussion

Current massive sequencing efforts yield unprecedented information regarding microbial community composition during infection, but fail to provide the necessary information required to functionally understand microbial behaviors in mixed species communities. Moreover, while the bulk assays most frequently utilized to study microbial interactions have provided insight into how bacterial species can influence community survival, metabolism, or virulence factor production (*Hotterbeekx et al., 2017*), we lack detailed information regarding how single-cells respond to the presence of another species at initial encounters. Here, we developed quantitative live-imaging methods to visualize the behaviors of two clinically important organisms and track their behavior over time. Through these studies, we uncovered microbial behaviors that could not have been predicted from bulk assays alone. We observed that *P. aeruginosa* is able to alter its behavior in the presence of *S. aureus* from a distance by responding to secreted factors. *P. aeruginosa* is then able to tune its motility patterns, adopting a behavior we refer to as 'exploratory motility'.

What interspecies factors do *P. aeruginosa* sense and respond to? Our data thus far reveal that *P. aeruginosa* can not only respond to secreted products from *S. aureus*, but also more broadly to a range of model bacteria and CF pathogens, both Gram-positive and negative. *P. aeruginosa* increased surface motility to varying extents depending on the species and strain utilized – a feature we predict will inform the nature and breadth of factors to which *P. aeruginosa* is capable of

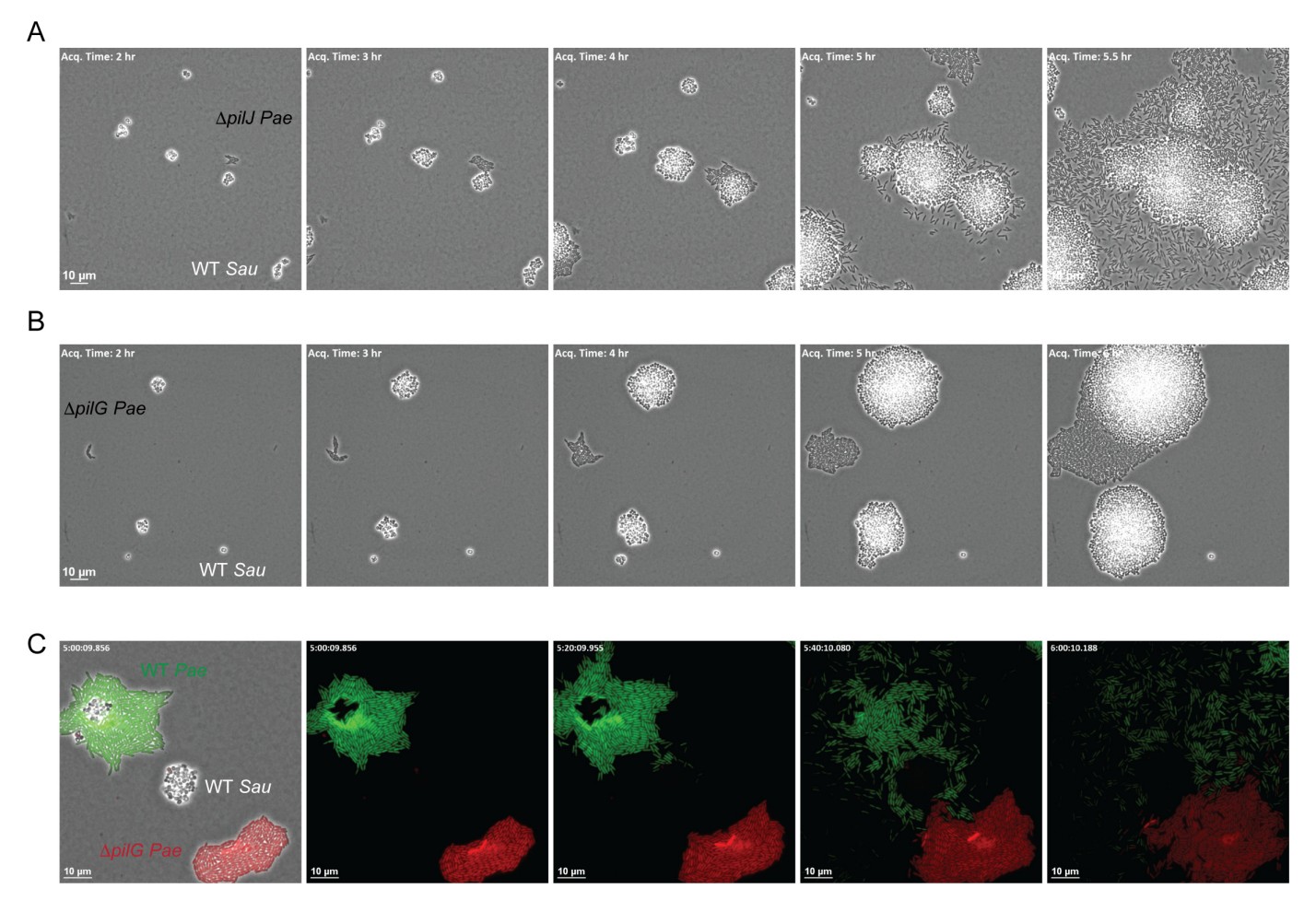

**Figure 6.** The CheY - like response regulator, PilG modulates *P. aeruginosa* response to *S. aureus*. Representative snap-shots of *P. aeruginosa* Δ*pilJ* (**A**) and Δ*pilG* (**B**) in coculture with WT *S. aureus*. In (**C**), representative snap-shots of WT *P. aeruginosa* (GFP, green) with *P. aeruginosa* Δ*pilG* (mKate, red) and WT *S. aureus* (unmarked), visible only in the first frame (left) with phase contrast overlay.

The online version of this article includes the following figure supplement(s) for figure 6:

**Figure supplement 1.** Motility of WT, Δ*pilA*, Δ*pilJ*, and Δ*pilG P. aeruginosa* by macroscopic sub-surface inoculation assays in the presence of medium alone (TSB) or cell-free supernatant from WT *S. aureus* strains.

**Figure supplement 2.** Live-imaging of PAO1 Δ*pilG* in coculture with WT PAO1 and WT *S. aureus*.

**Figure supplement 3.** Live-imaging of PA14 Δ*pilG* in coculture with increased *S. aureus* inoculum.

responding. For *S. aureus,* we determined the factors are heat stable proteins regulated by the Agr quorum sensing system. This insight led to the determination that *S. aureus* PSMs contribute to the ability of *S. aureus* to induce surface motility in *P. aeruginosa*, but do not fully account for the observed phenotype, suggesting that additional unidentified factors are necessary. Moreover, *S. aureus* may also produce Agr-independent factors that influence *P. aeruginosa* motility, given that an Agr-deficient strain is unable to induce motility in the macroscopic sub-surface inoculation motility assay or directional motility on either the macroscopic or single-cell level, but remains capable of increasing single-cell motility during live-imaging. Thus, it is formally possible that separate factors are required for initiating single-cell motility and directional motion. Live-imaging also suggests that *P. aeruginosa* cells may actively avoid Agr-deficient *S. aureus*, raising the possibility that the Agr system negatively regulates a secreted factor that is unfavorable for *P. aeruginosa*.

How do PSMs promote *P. aeruginosa* motility? PSMs are amphipathic peptides that are able to reduce interfacial tension (IFT), that is, the cohesive or excess energy between molecules at an interface. While such surfactants have been extensively characterized to play an essential role in flagella-

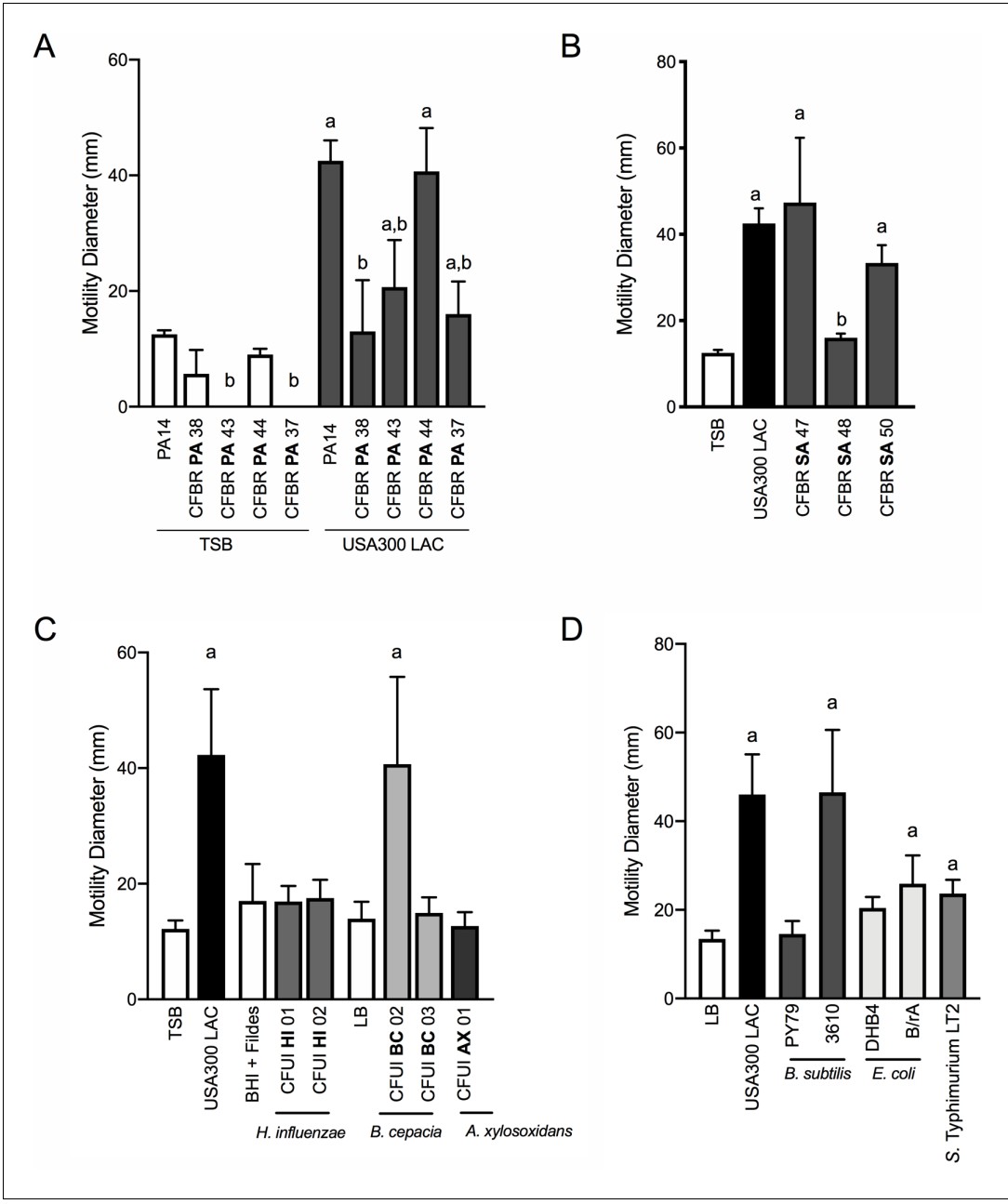

**Figure 7.** *P. aeruginosa* modulates motility in response to a range of CF clinical and non-clinical bacterial species. The motility of clinical *P. aeruginosa* isolates (CFBR PA 38, 43, 44, and 37), in the presence of cell-free supernatant derived from *S. aureus* USA300 LAC (0.5 dilution) is indicated, in comparison to *P. aeruginosa* PA14 (**A**). The motility of P. aeruginosa laboratory strain PA14 was monitored in the presence of undiluted cell-free supernatant from clinical *S. aureus* CF isolates, CFBR SA 47, 48, and 50 (**B**), CF clinical isolates: *H. influenzae, B. cepacia,* and *A. xylosoxidans* (**C**), and non-CF species: *E. coli, B. subtilis,* and *S.* Typhimurium (**D**). Growth medium alone for each species was used as a negative control (TSB: *S. aureus,* BHI + Fildes: *H. influenzae,* LB: *B. cepacia, A. xylosoxidans, B. subtilis, E. coli,* and *S.* Typhimurium). The mean and standard deviation are indicated for at least three biological replicates. Statistical significance was determined by one-way ANOVA followed by Tukey's Multiple Comparisons Test – *a* indicates a statistically significant difference (p≤0.05) between the motility observed in the presence of *S. aureus* supernatant compared to medium alone (**A–D**), and *b* indicates a statistically significant difference (p≤0.05) between the motility observed in the CF isolates, in comparison to laboratory strains (*P. aeruginosa* laboratory strain PA14 in (**A**) and *S. aureus* USA300 LAC in (**B**)).

The online version of this article includes the following figure supplement(s) for figure 7:

**Figure supplement 1.** Clinical CF *S. aureus* isolates retain β-hemolysis.

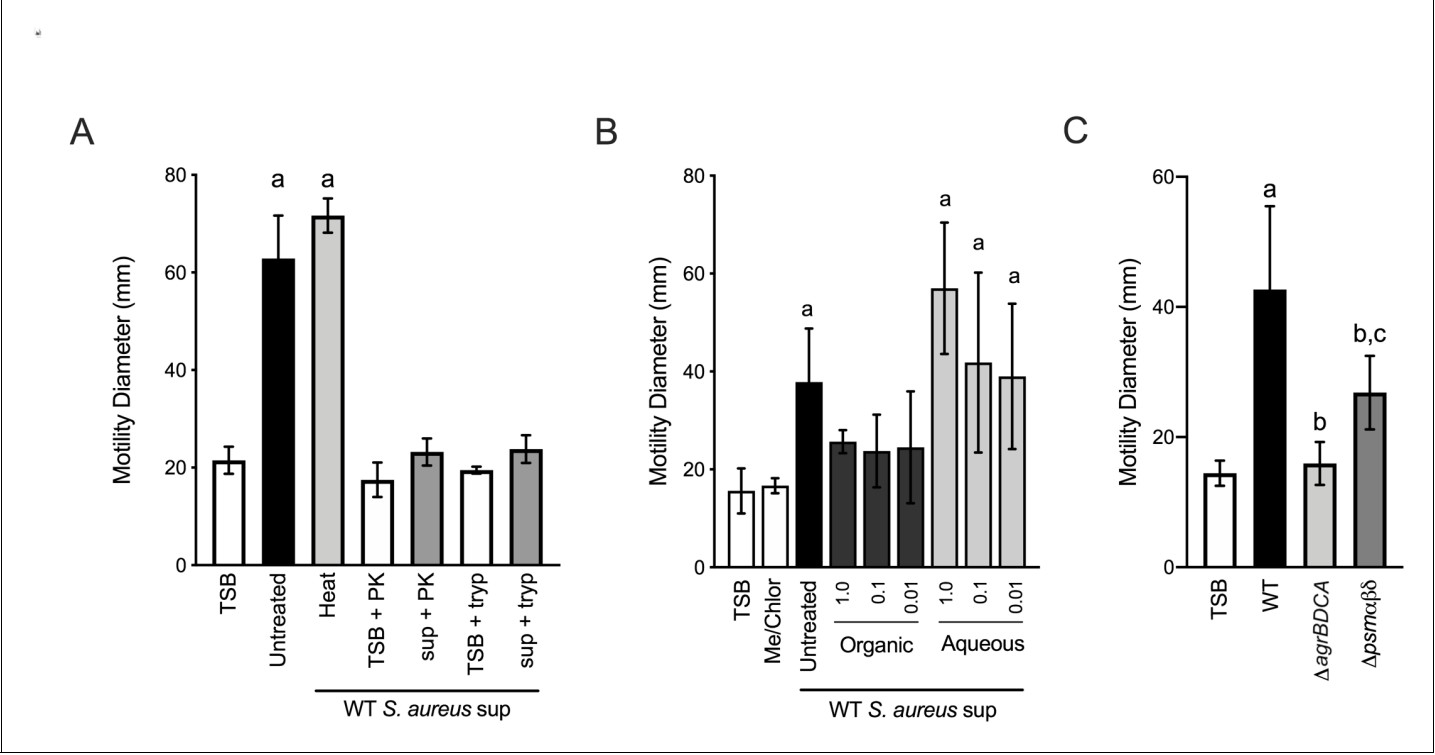

**Figure 8.** Phenol soluble modulins contribute to *S. aureus*-induced *P. aeruginosa* motility. Motility of WT *P. aeruginosa* was monitored by macroscopic sub-surface inoculation assays in the presence of medium alone or cell-free supernatant derived from WT *S. aureus* treated with either heat, proteinase K (PK), or trypsin (tryp) in (**A**), methanol/chloroform extraction of the supernatant and subsequent analysis of the organic and aqueous fraction (methanol-chloroform (2:1, Me/Chlor) was included as the vehicle control) (**B**), or in the presence of untreated supernatant derived from WT or the indicated *S. aureus* mutants (0.5 dilution) in (**C**). The mean and standard deviation are indicated for at least three biological replicates. Statistical significance was determined by one-way ANOVA followed by Tukey's Multiple Comparisons Test - *a* indicates a statistically significant difference (p≤0.05) between the motility observed in the presence of *S. aureus* supernatant compared to medium alone, *b* between the motility observed in the mutant strains compared to the parental, and *c* between the ΔagrBDCA and Δpsmαβδ mutants.

mediated bacterial surface motility, referred to as swarming (*Kearns, 2010*), their role in TFP-mediated motility is not well described. TFP generate bacterial movement on surfaces though polymerization and extension of pilus fibers, followed by pilus retraction, which pulls the cell body forward. Reduced IFT between the cell body and the surface, without affecting interaction between the pilus tip and the surface, may reduce drag on the cell body and reduce the force necessary to retract the pilus. TFP-mediated motility is most often described as a behavior where cells are primarily motile only in large groups. The behaviors for WT *P. aeruginosa* in monoculture observed here are consistent with previous descriptions of twitching motility, including the movement of cells in rafts, preferentially aligned along their long axis, and group tendril formation at the edges of the expanding community (*Burrows, 2012*). In contrast, in the presence of *S. aureus*, *P. aeruginosa* cells were seen to exit an expanding raft and transition to single-cell motility. Perhaps PSMs also reduce the IFT between neighboring *P. aeruginosa* cells, disrupting collective movement within rafts.

PSMs are known to promote *S. aureus* spreading on wet surfaces, in particular PSMα3 and δ-toxin, which possess high surfactant activity and low hydrophobicity (*Tsompanidou et al., 2013*). In the current study, we examined a mutant deficient in all seven PSMs, thus it is unknown if each *S. aureus* PSM possesses similar ability to promote motility and/or if their mechanism of action toward *P. aeruginosa* will also depend on their specific surfactant activity. Motility experiments in the presence of *B. subtilis* support the hypothesis that surfactants may increase TFP-mediated motility in *P. aeruginosa*. Two strains were examined, PY79 and 3610. 3610 induced *P. aeruginosa* motility similar to *S. aureus*. In comparison to PY79, which did not induce motility and is deficient in surfactant production, 3610 is an undomesticated strain that secretes a potent lipopeptide surfactant, called surfactin (*Kearns and Losick, 2003*). Thus, it is interesting to speculate that the different levels of

surfactin account for the differential ability of these *B. subtilis* strains to promote *P. aeruginosa* motility.

PSMs can also form small transient pores in lipid membranes resulting in cell lysis of eukaryotic and some prokaryotic cells (*Cheung et al., 2014*). Thus, it is possible that interactions of PSMs with the *P. aeruginosa* outer membrane influence *P. aeruginosa* surface motility, independent of reduced IFT at the surface. For example, surfactin has been shown to influence *B. subtilis* quorum sensing and biofilm formation through the formation of membrane pores, resulting in potassium leakage. The resulting low cellular potassium activates the protein kinase, KinC, which regulates biofilm formation (*López et al., 2009*). Future studies will investigate if PSMs influence *P. aeruginosa* surface motility by reduction of IFT, interactions with the *P. aeruginosa* outer membrane, and/or alternative mechanism(s). Future studies are also necessary to determine if PSMs are sufficient to promote directional *P. aeruginosa* motility or if PSMs function in concert with additional *S. aureus* factors to increase surface motility per se in order to facilitate movement towards an additional stimulus.

While flagella-mediated chemotaxis has been extensively studied in *P. aeruginosa*, little is understood regarding how bacteria direct movement on a surface utilizing the TFP. While the *P. aeruginosa* Pil-Chp system harbors similarity to the well-studied Che system in *E. coli*, initial studies uncover significant differences in the regulation of this pathway. PilJ is a predicted chemoreceptor, by sequence similarity to the flagella chemotaxis system (*Darzins, 1994*); however, the functional role in chemotaxis remains unclear. In initial studies of TFP-mediated chemotaxis, *P. aeruginosa* was shown to direct motility on agar surfaces up a gradient of phosphatidylethanolamine (PE) (*Kearns et al., 2001*). While a Δ*pilJ* mutant did not exhibit preferential movement up a PE gradient, the lack of twitching motility in this mutant prohibited interpretation of a role for PilJ. Here, despite the lack of development of a macroscopic twitching motility zone, individual cells were capable of initiating motility in the presence of *S. aureus* when monitored by live-imaging, suggesting that *P. aeruginosa* is capable of sensing *S. aureus* and performing twitching motility in the absence of functional PilJ.

How does *P. aeruginosa* sense an external stimulus and transmit this into a mechanical response driving twitching motility? Twitching cells move by pulling themselves along by pili clustered at one pole. Cells can reverse direction by extending the pili from the opposite pole, changing the direction of movement along the long axis. These reversals have recently been suggested to be important for *P. aeruginosa* to direct movement up a chemotactic gradient while on a surface (*Oliveira et al., 2016*). In a Δ*pilG* mutant (the response regulator for pilus extension), cells were observed to perform fewer reversals in comparison to WT in response to a chemotactic gradient and thus were unable to bias the directionality of their movement. Here we examined a role for PilG in modulating a response to secreted factors from *S. aureus* and found a Δ*pilG* mutant was significantly diminished in the ability to respond. Both Δ*pilJ* and Δ*pilG* mutants have been previously reported to be deficient in producing traditional twitch rings in the macroscopic sub-surface inoculation assay (*Luo et al., 2015*; *Bertrand et al., 2010*; *Darzins, 1994*). Our observations here for PilJ and PilG, combined with those from Oliveira et al. suggest the collective movements of these mutants as measured in bulk assays may not be reflected on the single-cell level. However, in comparison to studies by Oliveira, we observed very little movement for *P. aeruginosa* in the absence of PilG. These studies utilized the *P. aeruginosa* parental strain PAO1 (in comparison to PA14 used here). We hypothesized that differences in parental strain background (PAO1 verses PA14) may account for the differences observed; however, we acquired the PAO1 parental and PAO1 Δ*pilG* mutant utilized in these studies and obtained similar results as observed for PA14 in coculture with *S. aureus* (*Figure 6—figure supplement 2*). Therefore, it is likely that experimental conditions influence to what extent *P. aeruginosa* is capable of performing TFP-mediated motility in the absence of PilG.

*Myxococcus xanthus* is one of the most notorious predatory bacteria – elaborating an array of social behaviors reminiscent of what we observe here for *P. aeruginosa*. *M. xanthus* coordinates a cooperative, density-dependent feeding behavior, resulting in propulsion of the cells rapidly through the colony of prey, leading to prey lysis, and nutrient acquisition. When *M. xanthus* contacts prey cells, pili-dependent reversals are stimulated, which keeps the cells 'trapped' near the prey colony, promoting contact-dependent prey killing and increased local concentration of antimicrobials (*Muñoz-Dorado et al., 2016*). Similarly, we observed that when *P. aeruginosa* encounters *S. aureus*, the *P. aeruginosa* cells appear to mount a coordinated response whereby *P. aeruginosa* surrounds the *S. aureus* colony and eventually invades the colony – a behavior referred to as the 'wolf pack'

strategy. *P. aeruginosa* produces an arsenal of secreted antimicrobials shown to inhibit the growth of *S. aureus*, yet these secreted factors alone are insufficient for cellular lysis (*Limoli et al., 2017*). Thus, it is interesting to speculate that these behaviors function to synergistically increase cellular contacts necessary to kill *S. aureus* by a contact-dependent mechanism and/or to locally increase the concentration of secreted products.

By acquiring a fundamental understanding of how bacteria sense and respond to life with each other, we move closer to learning how to rationally manipulate interspecies behaviors during infection and in the environment. While for CF patients, this may mean preventing *P. aeruginosa* and *S. aureus* physical interactions, in other instances, we might bring species together who synergize to produce a beneficial compound.

# Materials and methods

**Key resources table**

| Reagent type (species) or resource | Designation | Source or reference | Identifiers | Additional information |
|---|---|---|---|---|
| Strain, strain background (*Escherichia coli*) | DH5α | Life Technologies | *supE44 ΔlacU169* (φ80d*lacZ*ΔM15) *hsdR*17 *thi-1 relA1 recA1* | |
| Strain, strain background (*Escherichia coli*) | S17 λpir | Life Technologies | *TpR SmR recA, thi, pro, hsdR*-M+RP4: 2 -Tc:Mu:Km Tn7 λpir* | |
| Strain, strain background (*Escherichia coli*) | B/rA | ATCC | ATCC 12407 | Obtained from David Weiss (Iowa) |
| Strain, strain background (*Escherichia coli*) | MC4100 | PMID: 11677609 | F− (*araD139*) Δ(*lac*) U169, strA, thi | Obtained from David Weiss (Iowa) |
| Strain, strain background (*Pseudomonas aeruginosa*) | PA14 (WT) | PMID: 7604262 | Non-mucoid prototroph SMC232 (O'Toole Strain Collection) | |
| Strain, strain background (*Pseudomonas aeruginosa*) | PA14 mKO | PMID: 28084994 | *attB*:: P$_{A1/04/03}$ -mKO Constitutively produces orange fluorescent protein | Obtained from Carey Nadell (Dartmouth) |
| Strain, strain background (*Pseudomonas aeruginosa*) | PA14 Δ*pilA* | PMID: 20233936 | SMC3782 (O'Toole Strain Collection) | |
| Strain, strain background (*Pseudomonas aeruginosa*) | PA14 Δ*flgK* | PMID: 28167523 | SMC5845 (O'Toole Strain Collection) | |
| Strain, strain background (*Pseudomonas aeruginosa*) | PA14 Δ*pilA* Δ*flgK* | PMID: 28167523 | SMC6595 (O'Toole Strain Collection) | |
| Strain, strain background (*Pseudomonas aeruginosa*) | PA14 Δ*pilJ* | PMID: 18178737 | SMC2992 (O'Toole Strain Collection) | |
| Strain, strain background (*Pseudomonas aeruginosa*) | PA14 Δ*pilG* | | SMC4375 (O'Toole Strain Collection) | Obtained from Kyle Cady (Dartmouth) |
| Strain, strain background (*Pseudomonas aeruginosa*) | PAO1 Δ*pilG* | PMID: 20008072 | | Obtained from Joanne Engel (UCSF) |

*Continued on next page*

*Continued*

| Reagent type (species) or resource | Designation | Source or reference | Identifiers | Additional information |
|---|---|---|---|---|
| Strain, strain background (*Pseudomonas aeruginosa*) | CFBR PA37 | PMID: 28325763 | Mucoid cystic fibrosis isolate | Obtained from the CF Biospecimen Registry (Emory) |
| Strain, strain background (*Pseudomonas aeruginosa*) | CFBR PA38 | PMID: 28325763 | Mucoid cystic fibrosis isolate | Obtained from the CF Biospecimen Registry (Emory) |
| Strain, strain background (*Pseudomonas aeruginosa*) | CFBR PA43 | PMID: 28325763 | Mucoid cystic fibrosis isolate | Obtained from the CF Biospecimen Registry (Emory) |
| Strain, strain background (*Pseudomonas aeruginosa*) | CFBR PA44 | PMID: 28325763 | Non-mucoid cystic fibrosis isolate | Obtained from the CF Biospecimen Registry (Emory) |
| Strain, strain background (*Staphylococcus aureus*) | USA300 LAC (WT) | PMID: 23404398 | USA300 CA-Methicillin resistant strain LAC without plasmids | Obtained from Ambrose Cheung (Dartmouth) |
| Strain, strain background (*Staphylococcus aureus*) | USA300 LAC pCM29 | PMID: 20829608 | *sar*AP1-sGFP | Obtained from Kenneth Bayles (Nebraska) |
| Strain, strain background (*Staphylococcus aureus*) | USA300 LAC *agrB*::TnMar | PMID: 23404398 | | Obtained from the Nebraska Transposon Mutant Library |
| Strain, strain background (*Staphylococcus aureus*) | USA300 LAC *sigB*::TnMar | PMID: 23404398 | | Obtained from the Nebraska Transposon Mutant Library |
| Strain, strain background (*Staphylococcus aureus*) | USA300 LAC Δ*agrBDCA* | This study | | See Materials and methods |
| Strain, strain background (*Staphylococcus aureus*) | USA300 LAC Δ*agr BDCA* + WT *agrBDCA* | This study | | See Materials and methods |
| Strain, strain background (*Staphylococcus aureus*) | CFBR SA47 | PMID: 28325763 | Cystic fibrosis isolate | Obtained from the Nebraska Transposon Mutant Library |
| Strain, strain background (*Staphylococcus aureus*) | CFBR SA48 | PMID: 28325763 | Cystic fibrosis isolate | Obtained from the CF Biospecimen Registry (Emory) |
| Strain, strain background (*Staphylococcus aureus*) | CFBR SA50 | PMID: 28325763 | Cystic fibrosis isolate | Obtained from the CF Biospecimen Registry (Emory) |
| Strain, strain background (*Haemophilus influenzae*) | CFUI HI01 | This study | Cystic fibrosis isolate | Obtained from U Iowa CF Biobank |
| Strain, strain background (*Haemophilus influenzae*) | CFUI HI02 | This study | Cystic fibrosis isolate | Obtained from U Iowa CF Biobank |

*Continued on next page*

*Continued*

| Reagent type (species) or resource | Designation | Source or reference | Identifiers | Additional information |
|---|---|---|---|---|
| Strain, strain background (*Burkholderia cepacia*) | CFUI BC02 | This study | Cystic fibrosis isolate | Obtained from U Iowa CF Biobank |
| Strain, strain background (*Burkholderia cepacia*) | CFUI BC03 | This study | Cystic fibrosis isolate | Obtained from U Iowa CF Biobank |
| Strain, strain background (*Achromobacter xylosoxidans*) | CFUI AX01 | This study | Cystic fibrosis isolate | Obtained from U Iowa CF Biobank |
| Strain, strain background (*Bacillus subtilis*) | PY79 | PMID: 6093169 | Prototrophic derivative of *B. subtilis* 168 | Obtained from Craig Ellermeier (Iowa) |
| Strain, strain background (*Bacillus subtilis*) | 3610 | PMID: 11572999 | NCIB 3610: Marburg 'wild' isolate | Obtained from Craig Ellermeier (Iowa) |
| Strain, strain background (*Salmonella enterica* serovar Typhimurium) | LT2 | PMID: 11677609 | | Obtained from Bradley Jones (Iowa) |
| Strain, strain background (*Staphylococcus aureus*) | USA300 LAC Δ*psmαβδ* | PMID: 26077761 | *psmΔα1–4 Δβ1–2 δ*ATG-ATT | Obtained from Daniel Wozniak (OSU) |
| Recombinant DNA reagent | pMAD | PMID: 15528558 | *E. coli-S. aureus* shuttle vector. OripE194$^{TS}$ *ermC blaC bgaB* | |
| Commercial assay or kit | Gibson Assembly Cloning Kit | New England BioLabs | E5510 | |
| Software, algorithm | NIS-Elements AR | Nikon | Version 5.2 | |

## Bacterial strains and culture conditions

*P. aeruginosa* and *E. coli* were routinely cultured in lysogeny broth (LB; 1% tryptone, 0.5% yeast extract, 1% sodium chloride) and *S. aureus* in tryptic soy broth (TSB, Becton Dickenson) at 37°C, with aeration. For coculture assays, both species were grown in TSB or M8 minimal medium (48 mM sodium phosphate dibasic, 22 mM potassium phosphate monobasic, 8.6 mM sodium chloride, 2.0 mM magnesium sulfate, 0.1 mM calcium chloride) supplemented with 1% glucose and tryptone. When necessary for strain construction, medium was supplemented with the following antibiotics: gentamicin (10 µg/ml *E. coli*; 30 µg/ml *P. aeruginosa*), ampicillin (100 µg/ml *E. coli*), carbenicillin (200 µg/ml *P. aeruginosa*), or chloramphenicol (10 µg/ml *S. aureus*).

*P. aeruginosa* and *S. aureus* clinical isolates were obtained from the CF Biospecimen Registry (CFBR) at the Children's Healthcare of Atlanta and Emory University CF Discovery Core and the *B. cepacia, A. xylosoxidans,* and *H. influenzae* isolates were obtained from the University of Iowa Cystic Fibrosis Biobank. *B cepacia, A. xylosoxidans, B. subtilis,* and *S.* Typhimurium were grown in LB and *H. influenzae* in Brain Heart Infusion (BHI) broth supplemented with 5% Fildes Enrichment at 37°C with aeration.

*S. aureus genetic manipulation*. In-frame deletion of the *agrBDCA* operon in a *S. aureus* USA300 LAC strain (JE2) was generated using pMAD-mediated allelic replacement (*Arnaud et al., 2004*). Briefly, a pMAD deletion vector was created by Gibson assembly (*Gibson et al., 2009*) using EcoRI/BamHI-linearized pMAD and PCR amplicons of 1 kb regions up- and downstream of *agrBDCA* (primer sets agr-a/agr-b and agr-c/agr-d, respectively, *Table 1*). The vector to chromosomally complement this deletion strain was similarly produced using Gibson assembly on the EcoRI/BamHI-linearized pMAD vector and the PCR product of the agr-a/agr-d primers. Following the standard

**Table 1.** Oligonucleotides used in this study.

| Name | Sequence | Reference |
|------|----------|-----------|
| agr-a | CGCGGATCCTACATAGCACTGAGTCCAAGG | This study |
| agr-b | GCCGTTAACTGACTTTATTATCTTTTTTACACCACTCTCCTCACTG | This study |
| agr-c | CAGTGAGGAGAGTGGTGTAAAAAAGATAATAAAGTCAGTTAACGGC | This study |
| agr-d | CCGGAATTCCAGTTATTAGCAGGATTTTAGC | This study |

protocol of heat shift on selective media, strains with successful deletions and restorations of *agrBDCA* were verified by PCR analysis and chromosomal DNA sequencing.

## Live-imaging of interspecies interactions

Bacteria were inoculated between a coverslip and an agarose pad and imaged with time-lapse microscopy. Pads were made by adding 900 µl of M8 medium with 1% glucose, 1% tryptone, and 2% molten agarose to a 10 mm diameter silicone mold on a coverslip. Pads were allowed to dry for 1 hr at room temperature, followed by 1 hr at 37°C. Meanwhile, bacteria were prepared by growing to mid-log phase in M8 medium with 1% glucose and 1% tryptone, diluting to $OD_{600}$ = 0.15 in warm media, and mixing *P. aeruginosa* and *S. aureus* 1:1. 2 µl of inoculum was spotted evenly onto the center of a warm 35 mm glass bottom dish, #1.5 mm coverglass (Cellvis). The agarose pad was removed from the mold and placed on top of the bacterial inoculum. Bacteria were immediately imaged with an inverted Nikon TiE or Ti2 at 37°C for 8 hr. Images were acquired with a 100x oil objective (1.45NA) with phase contrast and an ORCA Flash4.0 Digital CMOS camera (Hammamatsu).

Videos were generated and cells tracked and analyzed in Nikon Elements AR. For single-cells, binary images were generated and single *P. aeruginosa* cells were identified and tracking began when the first *P. aeruginosa* cell exited a raft. Rafts were tracked up until the time point where single-cell tracking began. Rafts edges were manually identified. The speed (µm/s), acceleration (µm/$s^2$), mean squared displacement (µm$^2$), accumulated track distance, $D_{(A)}$, and the Euclidean distance, $D_{(E)}$, were measured for at least 30 tracks (for single-cells) in four independent videos. The area of *S. aureus* was determined by dividing the total area occupied in a field of view divided by the number of colonies.

## Macroscopic coculture assays

*P. aeruginosa* sub-surface twitch (*Turnbull and Whitchurch, 2014*) and 0.3% soft agar motility assays (*Ha et al., 2014*) were performed as previously described with modifications for treatment with bacterial supernatant. Supernatant was derived by growing the indicated species overnight centrifuging to pellet cells, and supernatant filtered through a 0.22 µm filter. *S. aureus* was grown in TSB and strains normalized to $OD_{600}$ = 5.0 (and subsequently diluted to the indicated concentrations). For soft agar assays, cell-free supernatant was mixed at the indicated concentrations with cooled, but still molten agar prior to pouring. For subsurface inoculation assays, 100 µl of supernatant was spread onto the bottom of a petri plate before pouring media (1.5% agar). *P. aeruginosa* was grown to $OD_{600}$ = 2.0 in TSB and inoculated into motility plates by either stabbing with a toothpick halfway through the agar (0.3%), all the way though (sub-surface). Plates were incubated at 37°C for 24 hr, followed by 24 hr at room temperature. For twitch plates, agar was dropped out and the *P. aeruginosa* biomass stained with 1% crystal violet for visualization. The diameter of the motility zones was measured in mm.

## Biochemical analysis of *S. aureus* supernatant

Supernatant was collected as described above and exposed to the following conditions: 1. Heat: 95°C for 30 min. 2. Protease treatment: Proteinase K (400 µg/ml) at 55°C for 30 min or trypsin (2.5 µg/ml) at 37°C for four hours with gentle shaking, followed by heat inactivation at 95°C for 20 min. 3. Hydrophobic extraction: Performed according to the guidelines of *Bligh and Dyer (1959)*. In brief, a volume of 3.75 ml of methanol-chloroform (2:1) was added to 1.6 ml of *S. aureus* supernatant and vortexed for 1 hr. The suspension was centrifuged at 10,000 x *g* for 5 min and the aqueous phase

was saved. The organic phase was then reextracted with 4.75 ml of methanol-chloroform-water (2:1:0.8) and centrifuged again, and the extracted organic phases were combined.

### Directional twitching motility assay

Experiments were performed as previously described (*Miller et al., 2008*). In brief, buffered agar plates (10 mM Tris, pH 7.6; 8 mM $MgSO_4$; 1 mM $NaPO_4$, pH 7.6; and 1.5% agar) were poured and dried at room temperature for 24 hr. 4 μl of supernatant or medium (TSB) was spotted on the plates, and gradients were established by incubating the plates at 30°C for 24 hr. Supernatants were prepared as described above. *P. aeruginosa* strains were grown to early stationary phase, normalized to an $OD_{600}$ of 1.2 and 2 mls pelleted and resuspended in 100 ml of MOPS buffer (10 mM MOPS, pH 7.6, and 8 mM $MgSO_4$). 2 μl of this cell suspension was spotted approximately 6 mm from the center of the supernatant spot. Plates were incubated for 24 hr at 37°C followed by 24 hr at room temperature before images were taken of the twitching zones.

## Acknowledgements

This work was supported by funding from NIH Grant R37 AI83256 (GAO), CFF Postdoctoral Fellowship LIMOLI15F0, and CFF Postdoc-to-Faculty Transition Award LIMOLI18F5. We thank Drs. Ethan Garner, Gerard Wong, Jeffrey Meisner, and Minsu Kim for consultation on imaging studies, Kenneth Bayles, David Weiss, Craig Ellermeier, Bradley Jones, and Joanne Engel for bacterial strains, and Timothy Yahr for thoughtful discussions and careful reading of the manuscript.

## Additional information

### Funding

| Funder | Grant reference number | Author |
| --- | --- | --- |
| Cystic Fibrosis Foundation | Postdoctoral Fellowship LIMOLI15F0 | Dominique H Limoli |
| Cystic Fibrosis Foundation | CFF Postdoc-to-Faculty Transition Award LIMOLI18F5 | Dominique H Limoli |
| National Institutes of Health | Grant R37 AI83256 | George O'Toole |

The funders had no role in study design, data collection and interpretation, or the decision to submit the work for publication.

### Author contributions

Dominique H Limoli, Conceptualization, Data curation, Formal analysis, Funding acquisition, Investigation, Methodology, Writing—original draft, Writing—review and editing; Elizabeth A Warren, Data curation, Investigation, Methodology, Writing—review and editing; Kaitlin D Yarrington, Data curation, Formal analysis, Investigation, Writing—review and editing; Niles P Donegan, Resources, Investigation, Methodology, Writing—review and editing; Ambrose L Cheung, Resources, Supervision, Writing—review and editing; George A O'Toole, Conceptualization, Resources, Supervision, Funding acquisition, Writing—review and editing

### Author ORCIDs

Dominique H Limoli https://orcid.org/0000-0002-4130-337X
Kaitlin D Yarrington https://orcid.org/0000-0002-5347-2760
Niles P Donegan http://orcid.org/0000-0002-8328-2044

### Decision letter and Author response

Decision letter https://doi.org/10.7554/eLife.47365.sa1
Author response https://doi.org/10.7554/eLife.47365.sa2

## Additional files

### Supplementary files

• Source data 1. Data for macroscopic twitching motility assays.

• Transparent reporting form

### Data availability

All data generated or analyzed during this study are included in the manuscript and supporting files.

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
