## [Decision Letter]

**Acceptance summary:**

This work makes an important contribution to the literature by identifying a previously unrecognized interaction between two microbes that co-infect the CF airways: *Staphylococcus aureus* and *Pseudomonas aeruginosa*. Specifically, the authors show that *S. aureus* induces *P. aeruginosa* surface motility and identify phenol soluble modulins (together with yet-to-be identified other factors) as playing a role in inducing motility mediated by type IV pili. This work extends the relevance of single-cell twitching chemotaxis motility by *P. aeruginosa*, which had been previously reported, to interspecies interactions; it begins to uncover the mechanisms underpinning the interaction; and it demonstrates that other bacterial species can also induce *P. aeruginosa* motility, suggesting these findings may be generalizable. Together, it raises many interesting questions for future work, including whether similar factors produced by diverse species trigger this response, and whether evidence for this seemingly ecologically relevant in vitro interaction can be found in vivo. The authors are commended for being so responsive throughout the review process.

**Decision letter after peer review:**

Thank you for submitting your article "Interspecies signaling generates exploratory motility in *Pseudomonas aeruginosa*" for consideration by *eLife*. Your article has been reviewed by two peer reviewers, and the evaluation has been overseen by a Reviewing Editor and Gisela Storz as the Senior Editor. The following individual involved in review of your submission has agreed to reveal their identity: Matthew Wolfgang.

The reviewers have discussed the reviews with one another and the Reviewing Editor has drafted this decision to help you prepare a revised submission.

Summary:

The value of this paper is its focus on an interaction between 2 microbes that co-infect the CF airways. Though prior studies, such as Oliveira et al., 2016, have nicely documented single-cell twitching chemotaxis motility by *P. aeruginosa*, this work represents a significant step forward in relevance of this phenomenon to an important ecological context. We would like to see this study eventually published in *eLife*, pending a convincing response (involving both editorial changes as well as new experiments) to the following essential revisions.

Essential revisions:

The authors should do three things to improve the paper:

1) Better contextualize their study in light of the Oliveira paper (this can be done in the Introduction and Discussion, and also experimentally – see point 3).

2) Expand understanding of the specificity of this interspecies interaction: test whether *Pseudomonas* responds similarly to other non-Staph species and/or

3) Determine whether the interactions observed here are operating through the Chp chemotaxis genes shown by Oliveira to be important in *P. aeruginosa* in exploratory motility.

Reviewer #1:

In this study, Limoli et al. used single-cell imaging to discover a novel behavioral interaction between *Pseudomonas* aeruginosa and Staphylococcus aureus. By imaging cells in co-culture, they found that *P. aeruginosa* alters its motility in response to a chemical "signal" made by *S. aureus. P. aeruginosa* "senses" that signal and activates a form of motility that the authors term "exploratory motility." The authors go on to show that exploratory motility is driven, at least in part, by type IV pili. They also show that the signal generated by *S. aureus* is regulated by the Agr quorum sensing system. Overall, this work represents the beginning of a potentially exciting story with important implications to research on biofilms, bacterial interactions, and motility. However, in its current state the paper falls in between two incomplete stories. If the primary goal is to characterize a new behavioral transition of *P. aeruginosa* from collective to solitary movement, the authors need to better understand the mechanism underlying the different motility modes and the transition between them. And if the primary emphasis is the interspecies signaling, the authors need to better understand the nature of the signaling involved. In my view, answering one of these two questions would elevate the paper to the standards of *eLife*.

1) Is there really a "signal" sensed by *P. aeruginosa*? While it seems clear that *S. aureus* can affect *P. aeruginosa* motility and that its ability to do so depends on Agr, I am concerned that this could be a passive effect of a secreted factor (such as a surfactant or a protease) rather than an active signaling process. The argument for active changes in cAMP levels (Figure 6B) shows a modest decrease in cAMP levels when *P. aeruginosa* is exposed to *S. aureus*, but this effect is only seen at late time-points which do not appear to be consistent with the kinetics of the response (Figure 2A). To really confirm this point the authors would ideally be able to elicit the response with the purified signal. For example, if the signaling is direct, they could try simply adding AIP to *P. aeruginosa*. If not, the authors could better characterize the spent media activity they describe, for example by showing whether it is heat-stable, protease-resistant, organically-soluble, etc and then pursue purification appropriately.

2) How does the "signal" affect *P. aeruginosa* motility in general and type IV pili and flagella in particular? Figure 6 argues that cAMP drives exploratory motility, but I feel that this is simply correlation. In addition, it is unclear to me how changes in cAMP affect pilus activity and how pilus activity leads to "collective" or "single-cell" motility. At a minimum the authors could use single-cell imaging, perhaps with sparse labeling of fluorescent cells, to better understand the role of pili and flagella in collective motility. Additional cAMP mutants could help show the real necessity for this pathway. And finally, the role of PilJ needs to be untangled, as the signs of the signaling are confusing (how does inhibition of PilJ function stimulate solitary motility?).

Reviewer #2:

This manuscript By Limoli et al., describes a biologically interesting interspecies interaction between *P. aeruginosa* and *S. aureus* in vitro. Importantly, this is the first well-documented characterization of type IV pilus (Tfp) dependent chemotactic behavior in *P. aeruginosa*. Specifically, the authors demonstrate a bacterial behavior termed exploratory motility, in which individual *P. aeruginosa* cells actively move toward *S. aureus* colonies, and ultimately penetrate and disrupt the colony. The authors provide evidence that the chemotactic signal is secreted by *S. aureus* as 1) supernatants are sufficient to provoke the exploratory behavior and 2) the signal is lacking in *S. aureus* agr mutants.

The authors further confirm that the primary behavior requires *P. aeruginosa* Tfp, and that the phenotype can be modulated by altering intracellular cAMP (which is a regulator of Tfp biogenesis). While the data strongly suggests an underlying chemotactic response, the mechanisms is not thoroughly explored. Mutation of pilJ, encoding a putative MCP, resulted in an interesting phenotype, that was largely consistent with altered intracellular cAMP levels.

In summary, this is a very exciting paper describing an interesting and potentially significant biological behavior in an important opportunistic pathogen. The phenomenon is well documented, the experiments are well designed and the statistical analyses are thorough. I believe the manuscript will appeal to a broad audience and is likely to stimulate numerous additional studies.

I have only two major comments:

1) The authors propose that "exploratory behavior" may have implications for co-infections in cystic fibrosis and may represent a novel therapeutic target. This is highly speculative as the conditions in the CF lung are substantially different than the in vitro conditions used in the study. While *P. aeruginosa/S. aureus* co-infection is common in CF, *P. aeruginosa*a is an environmental bacterium that otherwise rarely coexists with *S. aureus*. It seems likely that this behavior evolved in an environmental niche, where it may provide a competitive advantage. It would be interesting to known if other species (in particular other environmental Gram-positive bacteria) can evoke the same phenotype in *P. aeruginosa*. This would provide information about the generality of the behavior in response to other potential bacterial competitors.

2) The signal and response mechanisms are only superficially explored. More information on either or both would increase the impact of the paper; however, I concede that an exhaustive characterization is unlikely to be achieved in a reasonable timeframe. Minimally, it would be useful to comment on other known factors implicated in *P. aeruginosa* Tfp function, including ChpA, PilG/PilH and PilY1. Also, a generic characterization of the chemotactic signal would be informative and support the findings; for example, is it a protein or heat labile.

[Editors' note: further revisions were requested prior to acceptance, as described below.]

Thank you for sending your article entitled "Interspecies signaling generates exploratory motility in *Pseudomonas* aeruginosa" for peer review at *eLife*.

While we appreciate that you successfully responded to our previous set of requests, and think this paper is on-track for acceptance in *eLife*, reviewer 1 raises an important point about the nature of the signal (and how it is discussed) that requires clarification. While we sympathize that revealing the identity of the signal might be something you are saving for another paper, inclusion of its identify here would significantly elevate the work. How easy would it be/how willing are you to include the identity of the unspecific "signal" in this manuscript, and/or test the hypothesis that it is surfactin (as suggested by reviewer 2)? If there is a compelling reason why taking it this one step further is impossible or undesirable, please explain.

Below, please find the direct critique from reviewer 1 that provides constructive feedback. Please address why you can or cannot comply with this request.

Reviewer #1:

In the original submission, Limoli et al. described an increase in motility of *P. aeruginosa* in the presence of *S. aureus*, which they argued could explain some of the interesting interactions known to occur between these species in clinically-important setting such as cystic fibrosis. Taking the two reviewers' comments into consideration, the editor requested that they address three major points: the species-specificity of the interaction, its genetic requirements, and its comparison to a previous potentially-related study by Oliveira et al.

The authors should be commended for indeed addressing all three points. Specifically, they now add additional data showing that the interactions are not species-specific and that the genetic requirements for their phenotype are similar to those previously reported by Oliveira et al.

On one hand, the authors successfully did what was asked of them. On the other hand, unfortunately, some of the new results dampened my enthusiasm for the work. For example, I am excited that clinical CF isolates of *P. aeruginosa* and *S. aureus* exhibit the same interspecies behavior as the model strains tested previously. However, I am worried that the new data showing that other bacterial species also induce *P. aeruginosa* motility indicates that the "signal" the authors are interested in is actually a more general secreted factor that passively induces motility. Specifically, I am intrigued that *B. subtilis* 3610 induces the response while *B. subtilis* PY79 (note: there is a typo in Figure 7D that calls it PV79) does not, as those two strains differ in their ability to make surfactin. Based on these new data, I feel that it is appropriate to directly test the hypothesis that surfactin is involved by testing a mutant of 3610 lacking the ability to make surfactin.

I would also encourage the authors to edit the tone of the Abstract and Introduction to better reflect their new findings. For example, I would feel more comfortable if the claim of "signaling" is decreased throughout the paper, as I am not convinced that active signaling is occurring. Along similar lines, the revised Abstract still makes it seem like the findings on the *P. aeruginosa-S. aureus* interactions are specific.

Reviewer #2:

The authors addressed my concerns in the revised manuscript. I have no additional comments or requests.

---

## [Author Response]

Essential revisions:The authors should do three things to improve the paper:1) Better contextualize their study in light of the Oliveira paper (this can be done in the Introduction and Discussion, and also experimentally – see point 3).

In response to both points 1 and 3, we further investigated a role for Chp chemotaxis genes in the context of the studies performed by Oliveira paper. This study described a role for the CheY-like response regulator PilG, which regulates the ATPase necessary for TFP extension (PilB), to play a role in *P. aeruginosa* response to a chemotactic gradient. We therefore examined a role for PilG in *P. aeruginosa* response to *S. aureus*. Similar to the results by Oliveira, we found that a Δ*pilG* mutant was diminished in its capacity to respond to *S. aureus* (Figure 6). This was particularly striking when we compared the motility of Δ*pilG* to Δ*pilJ*, the predicted TFP chemoreceptor. Despite exhibiting low motility in traditional macroscopic assays, the Δ*pilJ* mutant was capable of responding to *S. aureus*, while the response of the *pilG* mutant was significantly diminished. These data suggest that PilG participates in modulating directional TFP-mediated motility independent of the predicted chemoreceptor, PilJ.

These data are discussed in detail in the following areas of the edited manuscript:

Introduction; Results: subsection “The CheY-like response regulator, PilG, modulates *P. aeruginosa* response to *S. aureus”*; Figure 6 (including Figure 6—figure supplements 1-3) and Discussion.

2) Expand understanding of the specificity of this interspecies interaction: test whether Pseudomonas responds similarly to other non-Staph species and/or

In response to this helpful suggestion, we examined *P. aeruginosa* response to other non-Staph species, including clinical isolates from CF patients and also a range of model organisms, in order to determine the specificity of *P. aeruginosa* interactions with other bacterial species. We examined *Haemophilus influenzae, Burkholderia cepacia*, and *Achromobacter xylosoxidans* isolates from CF patients and *Bacillus subtilis, Escherichia coli*, and *Salmonella* Typhimurium laboratory strains. While the magnitude of the effect varied among organisms and strains, we found that *B. subtilis, E. coli, S. typhimurium*, and *B. cepacia* were capable of inducing motility in *P. aeruginosa*, suggesting induction of *P. aeruginosa* motility is not specific to *S. aureus* or CF pathogens. Investigation into the differences between the signals generated by these organisms is underway and is the focus of a follow-up manuscript.

These data are discussed in detail in the following areas of the edited manuscript:

Figure 7; Results: subsection “*P. aeruginosa* responds to a broad range of model organisms and CF pathogens” and Discussion.

3) Determine whether the interactions observed here are operating through the Chp chemotaxis genes shown by Oliveira to be important in P. aeruginosa in exploratory motility.

Please see response for point #1, above.

Reviewer #1:

In this study, Limoli et al. used single-cell imaging to discover a novel behavioral interaction between Pseudomonas aeruginosa and Staphylococcus aureus. By imaging cells in co-culture, they found that P. aeruginosa alters its motility in response to a chemical "signal" made by S. aureus. P. aeruginosa "senses" that signal and activates a form of motility that the authors term "exploratory motility." The authors go on to show that exploratory motility is driven, at least in part, by type IV pili. They also show that the signal generated by S. aureus is regulated by the Agr quorum sensing system. Overall, this work represents the beginning of a potentially exciting story with important implications to research on biofilms, bacterial interactions, and motility. However, in its current state the paper falls in between two incomplete stories. If the primary goal is to characterize a new behavioral transition of P. aeruginosa from collective to solitary movement, the authors need to better understand the mechanism underlying the different motility modes and the transition between them. And if the primary emphasis is the interspecies signaling, the authors need to better understand the nature of the signaling involved. In my view, answering one of these two questions would elevate the paper to the standards of eLife.1) Is there really a "signal" sensed by P. aeruginosa? While it seems clear that S. aureus can affect P. aeruginosa motility and that its ability to do so depends on Agr, I am concerned that this could be a passive effect of a secreted factor (such as a surfactant or a protease) rather than an active signaling process. The argument for active changes in cAMP levels (Figure 6B) shows a modest decrease in cAMP levels when P. aeruginosa is exposed to S. aureus, but this effect is only seen at late time-points which do not appear to be consistent with the kinetics of the response (Figure 2A). To really confirm this point the authors would ideally be able to elicit the response with the purified signal. For example, if the signaling is direct, they could try simply adding AIP to P. aeruginosa. If not, the authors could better characterize the spent media activity they describe, for example by showing whether it is heat-stable, protease-resistant, organically-soluble, etc and then pursue purification appropriately.

We thank the reviewer for these helpful comments. These questions are of significant interest to us as well and are the focus of current studies. We have evidence that suggests there are multiple signals, which can independently function to promote motility or function as a chemoattractant. Our preliminary studies indicate that the addition of *S. aureus* AIP is not sufficient to promote *P. aeruginosa* motility or chemotaxis. Thus, as the reviewer suggested, we have analyzed the spent supernatant and determined the factors necessary to increase motility are heat stable, protease sensitive, hydrophilic protein(s). We have identified candidate secreted *Staphylococcal* products we predict promote motility and/or chemotaxis and are working to define their specific roles therein. These experiments are the focus of a follow-up manuscript.

The reviewer brings up a second salient point regarding the kinetics of cAMP signaling in response to *S. aureus*. Experiments are currently underway to interrogate cAMP levels at the single cell level at earlier time points with live-imaging. We predict these experiments will also inform the important questions the reviewer raises in point 2, as we are able to more specifically visualize the kinetics of cAMP levels and the initiation of single-cell motility. We have therefore chosen to remove the current analysis and focused our discussion on the role of the PilG in light of the Oliveira paper, as recommended.

2) How does the "signal" affect P. aeruginosa motility in general and type IV pili and flagella in particular? Figure 6 argues that cAMP drives exploratory motility, but I feel that this is simply correlation. In addition, it is unclear to me how changes in cAMP affect pilus activity and how pilus activity leads to "collective" or "single-cell" motility. At a minimum the authors could use single-cell imaging, perhaps with sparse labeling of fluorescent cells, to better understand the role of pili and flagella in collective motility. Additional cAMP mutants could help show the real necessity for this pathway. And finally, the role of PilJ needs to be untangled, as the signs of the signaling are confusing (how does inhibition of PilJ function stimulate solitary motility?).

We agree with the reviewer that it is important to understand how changes in cAMP and pilus activity influence the transition between collective and single-cell motility. Studies are currently underway to measure pili biogenesis and activity via immunoblot and live-imaging of fluorescently labeled pili in the presence and absence of both *S. aureus* cells and secreted factors identified to be necessary as discussed above.

Please see response to point #1 above, and response to essential revisions #1.

Reviewer #2:

[…] I have only two major comments:

*1) The authors propose that "exploratory behavior" may have implications for co-infections in cystic fibrosis and may represent a novel therapeutic target. This is highly speculative as the conditions in the CF lung are substantially different than the* in vitro *conditions used in the study. While P. aeruginosa/S. aureus co-infection is common in CF, P. aeruginosaa is an environmental bacterium that otherwise rarely coexists with S. aureus. It seems likely that this behavior evolved in an environmental niche, where it may provide a competitive advantage. It would be interesting to known if other species (in particular other environmental Gram-positive bacteria) can evoke the same phenotype in P. aeruginosa. This would provide information about the generality of the behavior in response to other potential bacterial competitors.*

We thank the reviewer for these insightful comments. We absolutely agree that these interspecies behaviors likely evolved in an environmental niche, not the CF airway. Accordingly, we examined additional bacterial species, both isolates from CF patients and model organisms not common to the CF airway. As the reviewer suspected, *P. aeruginosa* was able to respond to a range of organisms, although some specificity was observed (please see response to essential revision #2 for more detail).

2) The signal and response mechanisms are only superficially explored. More information on either or both would increase the impact of the paper; however, I concede that an exhaustive characterization is unlikely to be achieved in a reasonable timeframe. Minimally, it would be useful to comment on other known factors implicated in P. aeruginosa Tfp function, including ChpA, PilG/PilH and PilY1. Also, a generic characterization of the chemotactic signal would be informative and support the findings; for example, is it a protein or heat labile.

These questions are of significant interest to us as well and are the focus of current studies. We have evidence that suggests there are multiple signals, which can independently function to promote motility or function as a chemoattractant. Thus, as the reviewer suggested, we have analyzed the spent supernatant and determined the factors necessary to increase motility are heat stable, protease sensitive, hydrophilic protein(s). We have identified candidate secreted *Staphylococcal* products we predict promote motility and/or chemotaxis and are working to define their specific roles therein. These experiments are the focus of a follow-up manuscript.

In response to comments by both reviewers, we have further investigated the role for PilG in light of the Oliveira paper. Please see response above to essential revision #1.

[Editors' note: further revisions were requested prior to acceptance, as described below.]

While we appreciate that you successfully responded to our previous set of requests, and think this paper is on-track for acceptance in eLife, reviewer 1 raises an important point about the nature of the signal (and how it is discussed) that requires clarification. While we sympathize that revealing the identity of the signal might be something you are saving for another paper, inclusion of its identify here would significantly elevate the work. How easy would it be/how willing are you to include the identity of the unspecific "signal" in this manuscript, and/or test the hypothesis that it is surfactin (as suggested by reviewer 1)? If there is a compelling reason why taking it this one step further is impossible or undesirable, please explain.

In response to the suggestion by reviewer #1 and recommendations from the editors, regarding the investigation of a role for surfactants in the induction of *P. aeruginosa* motility by *S. aureus,* we have included data describing the biochemical analysis of *S. aureus* supernatant we performed in an effort to identify the factor(s) made by *S. aureus* that promote motility in *P. aeruginosa*. These data reveal that the factors are heat stable, hydrophilic, and protease sensitive.

Since we previously demonstrated that Agr was necessary for the secretion of these products, we were able to identify *S. aureus* phenol soluble modulins (PSMs) as a putative factor necessary for induction of motility. We have included the genetic data showing that PSMs indeed play a role in the induction of *P. aeruginosa* motility, but additional unidentified factors are also likely required based on the data we present here. PSMs are multifunctional peptides: they can function as surfactants, lyse eukaryotic and some prokaryotic cells, and are chemoattractant for neutrophils. We speculate that the ability of PSMs to reduce interfacial tension increases *P. aeruginosa* TFP-mediated motility. As reviewer 1 notes, the observation that the undomesticated *B. subtilis* strain, known to secrete high levels of surfactant, also promotes *P. aeruginosa* motility supports this hypothesis. As we previously mentioned, experiments to understand how bacterial surfactants (including PSMs, surfactin, rhamnolipids, and serrawettin) promote T4P motility per se and influence the directionality of *P. aeruginosa* motility are underway. We have included a discussion of potential models in the manuscripts, but we feel that additional experimental investigation is outside the scope of this paper.

Below, please find the direct critique from reviewer 1 that provides constructive feedback. Please address why you can or cannot comply with this request.Reviewer #1:In the original submission, Limoli et al. described an increase in motility of P. aeruginosa in the presence of S. aureus, which they argued could explain some of the interesting interactions known to occur between these species in clinically-important setting such as cystic fibrosis. Taking the two reviewers' comments into consideration, the editor requested that they address three major points: the species-specificity of the interaction, its genetic requirements, and its comparison to a previous potentially-related study by Oliveira et al.The authors should be commended for indeed addressing all three points. Specifically, they now add additional data showing that the interactions are not species-specific and that the genetic requirements for their phenotype are similar to those previously reported by Oliveira et al.On one hand, the authors successfully did what was asked of them. On the other hand, unfortunately, some of the new results dampened my enthusiasm for the work. For example, I am excited that clinical CF isolates of P. aeruginosa and S. aureus exhibit the same interspecies behavior as the model strains tested previously. However, I am worried that the new data showing that other bacterial species also induce P. aeruginosa motility indicates that the "signal" the authors are interested in is actually a more general secreted factor that passively induces motility. Specifically, I am intrigued that B. subtilis 3610 induces the response while B. subtilis PY79 (note: there is a typo in Figure 7D that calls it PV79) does not, as those two strains differ in their ability to make surfactin. Based on these new data, I feel that it is appropriate to directly test the hypothesis that surfactin is involved by testing a mutant of 3610 lacking the ability to make surfactin.

Please see above for general response to this comment. Data describing these results are included in Figure 8 and in the Results. Further discussion is included in the Discussion section.

The typo in Figure 7D has been corrected.I would also encourage the authors to edit the tone of the Abstract and Introduction to better reflect their new findings. For example, I would feel more comfortable if the claim of "signaling" is decreased throughout the paper, as I am not convinced that active signaling is occurring. Along similar lines, the revised Abstract still makes it seem like the findings on the P. aeruginosa-S. aureus interactions are specific.

The use of the word signaling has been decreased throughout the paper, including the title and Abstract.